# High performance temperature difference triboelectric nanogenerator

Bolang Cheng[1,3], Qi Xu [1,3], Yaqin Ding[1,3], Suo Bai[1], Xiaofeng Jia[1], Yangdianchen Yu[1,2], Juan Wen [1✉] & Yong Qin [1✉]

Usually, high temperature decreases the output performance of triboelectric nanogenerator because of the dissipation of triboelectric charges through the thermionic emission. Here, a temperature difference triboelectric nanogenerator is designed and fabricated to enhance the electrical output performance in high temperature environment. As the hotter friction layer's temperature of nanogenerator is 0 K to 145 K higher than the cooler part's temperature, the output voltage, current, surface charge density and output power are increased 2.7, 2.2, 3.0 and 2.9 times, respectively (from 315 V, 9.1 μA, 19.6 μC m$^{-2}$, 69 μW to 858 V, 20 μA, 58.8 μC m$^{-2}$, 206.7 μW). With the further increase of temperature difference from 145 K to 219 K, the surface charge density and output performance gradually decrease. At the optimal temperature difference (145 K), the largest output current density is 443 μA cm$^{-2}$, which is 26.6% larger than the reported record value (350 μA cm$^{-2}$).

[1] Institute of Nanoscience and Nanotechnology, School of Materials and Energy, Lanzhou University, Lanzhou, China. [2] Department of Material Science and Engineering, College of Engineering, Boston University, Boston, MA, USA. [3] These authors contributed equally: Bolang Cheng, Qi Xu, Yaqin Ding. ✉email: wenj@lzu.edu.cn; qinyong@lzu.edu.cn

B ased on the coupling effect of the contact electrification and electrostatic induction, triboelectric nanogenerators (TENG) have been successfully developed for mechanical energy harvesting and self-powered sensors[1–3]. To obtain higher electrical output performance, lots of works have been done including material selection[4–6], structural optimization[7–10], artificial prior-charge injection[11,12], coupling surface polarization and hysteretic dielectric polarization in vacuum[13]. The relationships between output performance and materials, structure, friction areas, surface charge density, external forces, working frequency, etc. have also been studied[14–16]. Recently, current densities of 25.2 μA cm⁻² and 28.8 μA cm⁻² in the atmosphere have been obtained by a pumping TENG[17] and a self-charge excitation TENG system[18], respectively. The output can be improved to 57 μA cm⁻² by operating TENG in a vacuum environment[13] and to 90 μA cm⁻² by artificial prior-charge injection[11]. With an electric double-layer structure designed in TENG, 305 μA cm⁻² output can be obtained[10]. Moreover, by adopting ferroelectric materials and modulating dielectric permittivity to easier gain/loss charges simultaneously, the largest current density has been improved to 350 μA cm⁻² (ref. [19]).

Apart from room temperature, TENGs need to work at high temperatures in some particular applications, such as harvesting the vibration energy of automobile engines. But when the working temperature of TENG is higher than 260 K, its output decreases significantly[20,21], because the temperature of friction material will affect the storage and dissipation of electrons during triboelectrification[22,23]. In recent years, the electron thermionic emission in TENG has been studied[22–24], which is used to explain the influence of temperature on the output performance of TENG. It is found that the increase of the friction layer's temperature will increase the electron thermionic emission ability of the friction layer, thus reducing the charge storage ability of the friction layer, resulting in a poor output of TENG. So the temperature difference between two friction layers is a quite important and complicated influence factor on TENG's output. Later, keeping the temperature of the cooler end-tip constant and increasing the temperature of the dielectric sample's surface, the influence of the temperature difference on the nanoscale electron transfer process is studied, which reveals that the temperature difference will facilitate electrons transferring from hotter materials to cooler materials, and the charge density can be improved notably in a particular temperature difference range[24]. Although this work is done in nanoscale by using an atomic force microscope (AFM) tip, it gives a possibility in principle that the temperature difference between two friction layers can be utilized to enhance the output of TENG through rational design. In addition, temperature difference widely exists in some special conditions, such as the surface of airplane's engines and the exhaust pipe of automobiles. If both temperature difference energy and vibration energy of automobiles can be effectively converted into electricity, it is very beneficial for the comprehensive monitoring of the work conditions of airplanes and automobiles, car networking, etc. Moreover, it can also provide a way for high-efficiently energy harvesting of the temperature difference and vibration energy. However, in practical conditions, the temperature of the cooler friction layer will also rise due to the heat exchange between friction layers in contact-separation processes and air heat transfer, which will decrease the TENG's output performance. So, it is very important to enhance the output of TENG by utilizing the high temperature and temperature difference in practical application conditions, but it is still a great challenge now.

In this work, the influences of high temperature and temperature difference on TENG are studied theoretically, and the optimum temperature difference between friction layers for improving TENG's output is given. A temperature difference TENG with controllable friction layer temperature (TDNG) is successfully designed and fabricated to enhance the output

performance of TENG to a higher record. As the temperature difference between hotter and cooler friction layers ($\Delta T$) increasing from 0 K to 219 K, the electrical output performance of TDNG increases at first and then decreases. Under the optimal $\Delta T$ (~145 K), the open-circuit voltage of 858 V, short-circuit current of 20 μA, surface charge density of 58.8 μC m⁻², and output power of 206.7 μW are 2.7, 2.2, 3.0 and 4.9 times the output values when $\Delta T$ equals to 0 K, respectively. Furthermore, by further optimizing the friction materials of the TDNG, the current density is enhanced to 443 μA cm⁻², which is 26.6% larger than the record value (350 μA cm⁻²).

## Results

**Theoretical study of the influence of $\Delta T$ on TENG's output.** Taking the heat exchanges between hotter and cooler friction layers into account, the temperature of the cooler friction layer will continuously rise through the air and contacting heat transfer, and the accumulated charges in the cooler friction layer will gradually escape to the air as well as the hotter friction layer by electron thermionic emission. Here, an electron-cloud-potential-well model as shown in Fig. 1a is used to explain the electrons transfer process for triboelectrification between hotter and cooler friction layers when a temperature difference exists between the hotter and cooler frication layers. As the left part of Fig. 1a shows, the hotter friction layer's electron energy levels will increase ($\approx k\Delta T$) due to the raised temperature. On the one hand, during friction, electrons locating at a high energy level in the hotter friction layer will transit to the surface states of the hotter friction layer, and more electrons will hop from the hotter friction layer to the cooler friction layer (middle part of Fig. 1a) to enhance the charge density. On the other hand, because of the electron thermionic emission effect of triboelectric charges, as the right part of Fig. 1a shows, electrons are easier to escape out of the potential well and get back to the hotter friction layer in contact processes or spill into the air when the temperature of the cooler friction layer increases, which reduces charge density and output performance of TENG. As a result, the competition of the increased transferring charge and friction charge dissipation, results in a complex changing trend of the surface charge density. Therefore, there should have an optimal temperature difference that can boost the output performance of TENG.

Numerical simulations are carried out using COMSOL (Supplementary Fig. 1) to verify the above conjecture. During the simulation, electrons transferred from the hotter friction layer to the cooler friction layer as well as electrons dissipation from the cooler friction layer due to electron thermionic emission are considered simultaneously. In the first stage, as the inset of Fig. 1b shows, the temperature of the cooler friction layer ($T_c$) increases approximately linearly with the temperature of the hotter friction layer ($T_h$). In the second stage, according to the empirical law between charge density and $T_h$ (Eq. (1))[24], and the modified electron thermionic emission model (Eq. (2))[23]:

$$\sigma = -C_1 T_h + C_2 \tag{1}$$

$$\sigma_{tc} = e^{-SAt_0} \sigma_{tc0} \tag{2}$$

where $\sigma$ is the surface charge density when the temperature of the cooler friction layer keeps fixed, $C_1$ and $C_2$ are the material-related correction factors, $\sigma_{tc}$ is the short-circuit transfer charge density, and $\sigma_{tc0}$ is the initial value of $\sigma_{tc}$ (equal the value of $\sigma$), $A$ is the surface area of TENG, $t_0$ is the time of heat preservation, $S = \frac{\lambda_1 A_0}{k} T_c e^{\frac{qv}{kT_c}}$, where $\lambda_1$ is the material-specific correction factor, $A_0$ is Richardson constant of a free electron, $k$ is the Boltzmann constant, the relationship between surface charge density of TENG and temperature difference $\Delta T$ ($\Delta T = T_h - T_c$) is calculated (Fig. 1b). Unlike that in an ideally nanoscale

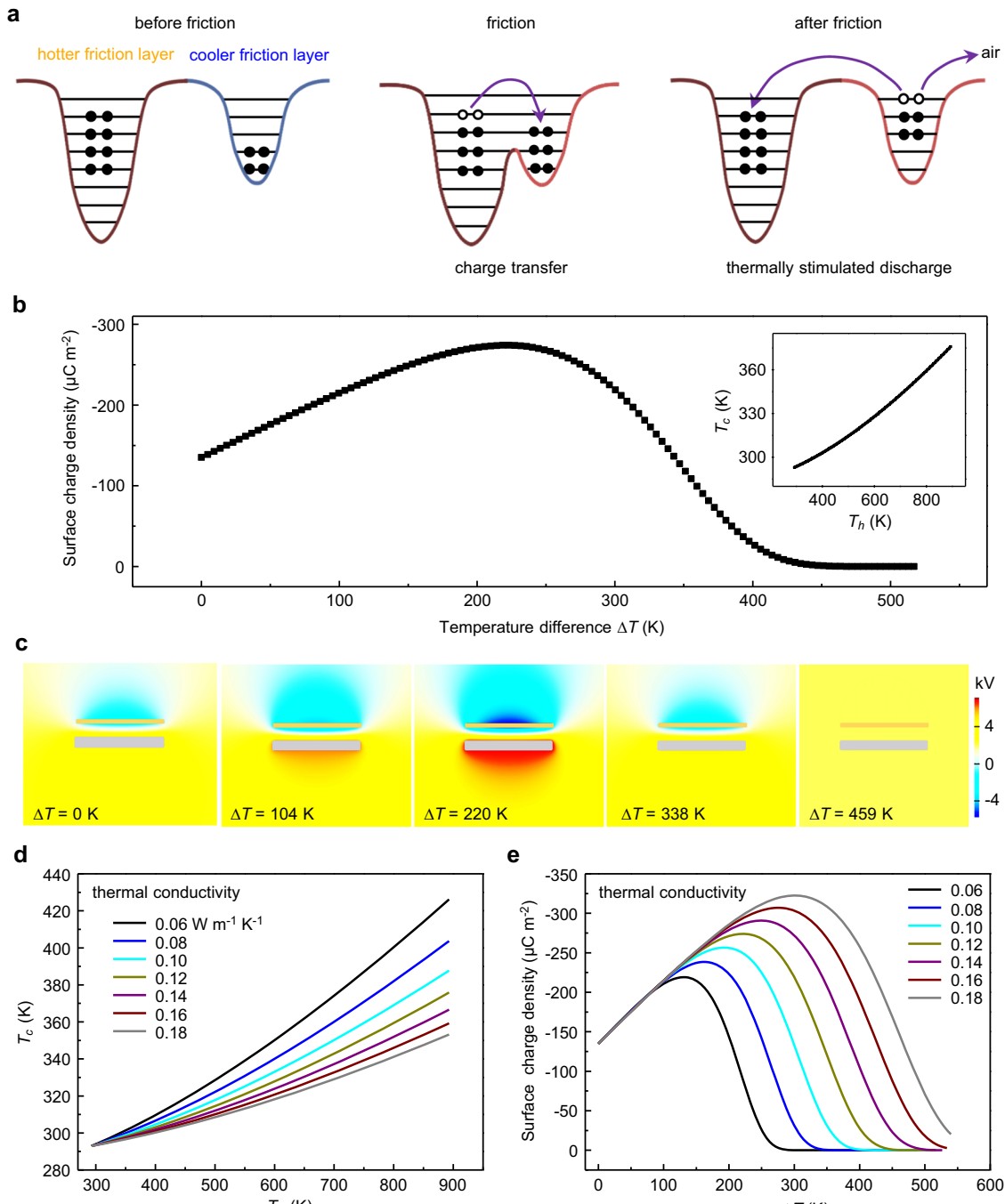

**Fig. 1 The influence of the temperature difference between different friction layers $\Delta T$ on the performance of triboelectric nanogenerator (TENG).**
**a** Effect of $\Delta T$ on charge transfer and dissipation. The black dots represent the electrons, and the black lines represent different energy levels. The brown, blue, and red curve represent the potential wells of different friction layers, respectively. **b** Numerical simulations of the relationship between $\Delta T$ and the short-circuit transfer charge density. The relationship between the temperature of the hotter friction layer ($T_h$) and the temperature of the cooler friction layer ($T_c$) is shown in the inset. **c** The potential distribution of TENG under different $\Delta T$. **d** The influence of the thermal conductivity of the cooler friction layer on $T_c$. **e** The influence of the thermal conductivity of the cooler friction layer on the surface charge density of the cooler friction layer.

situation (linearly increased transferred charges density with $\Delta T$)[24], the transferred charges density and surface potential (Fig. 1c) of TENG will increase and then decrease when $\Delta T$ increases to a certain level (above 220 K in simulation).

By combining Eq. (1) and Eq. (2), the overall influences of high temperature and low temperature on the output of TDNG can be described, and the factors favorable and adverse to surface charge density of TDNG are accounted together. The relationship between the surface charge density of TDNG ($\sigma_{TDNG}$) and $\Delta T$ can be described as follows

$$\sigma_{TDNG} = \left( -C_1 \frac{\Delta T + b}{1-a} + C_2 \right) e^{-SAt_0} \qquad (3)$$

where $S = \frac{\lambda_1 A_0 (a\Delta T + b)}{k(1-a)} e^{\frac{qv(1-a)}{k(a\Delta T + b)}}$, $a$ and $b$ are material and heat conduction related correction factors, which are related to the

thermal conductivity of friction materials. As shown in Fig. 1d, e, when the thermal conductivity of the cooler friction layer increases from 0.06 W m$^{-1}$ K$^{-1}$ to 0.18 W m$^{-1}$ K$^{-1}$, the effect of $T_h$ on $T_c$ gradually decreases (Supplementary Fig. 2), and the material-related correction factor $a$ change from 0.224 to 0.101. As shown in Fig. 1e, when the thermal conductivity of the cooler friction layer increases, the surface charge density at different $T_h$ and different $T_c$ but the same $\Delta T$ can be obtained. At the same $\Delta T$, the output of TDNG is related to the thermal conductivity of the materials, the larger thermal conductivity is, the higher the optimal temperature difference is, and the larger the surface charge density is. Through the classical electrodynamics derivation and by ignoring the effect of borders, the relationship between the voltage output of TDNG ($V_{(t)}$) and $\Delta T$ can be established as follows (the detailed derivation is given in Supplementary Note 1)

$$V_{(t)} = \frac{(-C_1 \frac{\Delta T + b}{1-a} + C_2)de^{-SAt_0}}{\varepsilon_0 \varepsilon_r}\left[ (d + x_{(t)}\varepsilon_r)\left(\frac{1}{d} + \frac{\int_0^t e^{\frac{1}{RA\varepsilon_0}\left(\frac{d}{\varepsilon_r}t + \int_0^t x_{(t)}dt\right)}dt}{RA\varepsilon_0\varepsilon_r}\right)e^{-\frac{1}{RA\varepsilon_0}\left(\frac{d}{\varepsilon_r}t + \int_0^t x_{(t)}dt\right)} - 1\right] \tag{4}$$

where $d$ is the thickness of the cooler friction layer, $\varepsilon_r$ is the permittivity of the cooler friction layer, $\varepsilon_0$ is the permittivity of the vacuum, $t$ is the time, and $R$ is the external resistance, $x_{(t)}$ is the distance between two friction layers, which is a function of time. Accordingly, if TENG works with a suitable temperature difference between friction layers, it can output a maximum electrical performance due to the enhanced surface charge density and higher transferred charges density.

**Fabrication and electrical output performance of TDNG.** Based on the theoretical analysis mentioned above, a TDNG includes a hotter part and a cooler part with a separation of the air gap (as shown in Fig. 2a) is designed and fabricated. Both the hotter part and cooler part including three layers: friction layer, electrode film, and temperature controllable heating/cooling system. Compared with previous TENGs, such a design can effectively control the temperature of friction layers, and it is convenient to study the influence of temperature difference on the output of TDNG. The friction layer in the hotter part and cooler part are the chemical reactive etched 20-μm-thick aluminum (Al) foil and directly reactive ion etched (RIE) 100-μm-thick Kapton film, respectively (the insets of Fig. 2a and Supplementary Fig. 3). The width and height of Al nanostructure are about 100 nm, and the width and height of these irregularly Kapton nanopillars are about 150 nm and 100 nm, respectively. The nanostructures fabricated on them can increase the surface area (1.87 times and 1.96 times of the untreated Al foil and Kapton film respectively, Supplementary Fig. 4) to enhance the effective friction area and improve TDNG's electrical output performance. In experiments, the temperature controllable heating and cooling system are composed of a thermostat heater and a water-cooling system, respectively (Supplementary Fig. 5).

To study the relationship between the output performance of TDNG and $\Delta T$, a linear motor (K15-W/C-2, LinMot, Supplementary Movie 1) is used to drive the TDNG. The working mechanism of TDNG is shown in Supplementary Fig. 6. Under a driving frequency of 0.7 Hz, the open-circuit voltage and short-circuit current at different $\Delta T$ are shown in Fig. 2b, c. In accordance with the simulation, with the $\Delta T$ increasing from 0 K to 219 K, the output voltage and current of TDNG increase at first and then decrease. The largest output voltage and current can reach 858 V (measured by SR560 and a voltage divider, Supplementary Fig. 7) and 20 μA when $\Delta T$ equals to 145 K, as $T_c$ keeps 299 K. Under this optimal $\Delta T$, the output voltage and current are 2.7 and 2.2 times the values when $\Delta T$ equals to 0 K

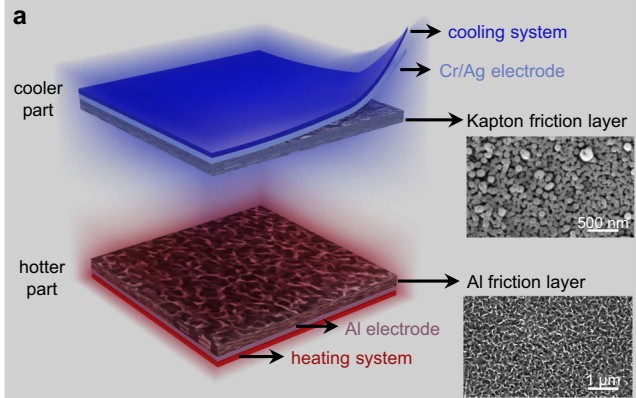

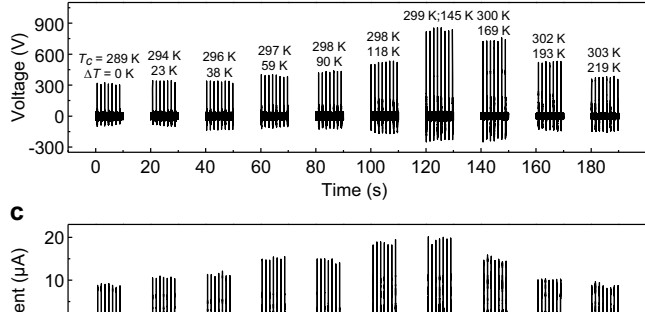

**Fig. 2 Design and output performance of temperature difference TENG with controllable friction layer temperature (TDNG). a** Schematic diagram of TDNG. Insets are scanning electron microscope images of nanostructures on Kapton (scale bar: 500 nm) and Al (scale bar: 1 μm). **b** The open-circuit voltage of TDNG under different $\Delta T$. **c** The short-circuit current of TDNG under different $\Delta T$.

(314.5 V, 8.98 μA). Besides, to avoid the influence of electrostatic induction, this phenomenon has also been verified under carbon heating condition (Supplementary Fig. 8).

To further study the effect of $\Delta T$ on TDNG's performance, the transferred charge quantities per cycle (CQC) of TDNG under different $\Delta T$ at short-circuit condition have been calculated and illustrated in Fig. 3a. Under the optimal $\Delta T$ (145 K), the CQC is 147 nC (corresponding to a surface charge density of 58.8 μC m$^{-2}$, Supplementary Fig. 9), which is 3 times the CQC when $\Delta T$ equals to 0 K at room temperature. Based on Eq. (3), a fitting analysis of CQC under different $\Delta T$ is performed. As shown in Supplementary Fig. 10, the obtained material-related correction factor $a$ equals to 0.1253, $b$ equals to 163.957, $C_1$ equals to 0.0248, and $C_2$ equals to 3.1166. From these fitting parameters, the thermal conductivity of the cooler friction layer can be deduced as 0.14 W m$^{-1}$ K$^{-1}$, which is approximately the same as the thermal conductivity of Kapton (~0.12 W m$^{-1}$ K$^{-1}$) that we used in experiments. In addition, to give a clearer illustration of the importance of temperature difference on TDNG, a thermally stimulated discharge (TSD) current is carried out to evaluate the amount of accumulated charges on Kapton (the accumulated charges in the friction layer is positively related to the surface charge density). As the schematic diagram of the TSD testing device (Supplementary Fig. 11) illustrates, when a charged Kapton film is heated by a heating system (the temperature rising curves of heating system in the TSD testing is shown in Supplementary Fig. 12), the accumulated charges in Kapton will gradually escape from the potential well and

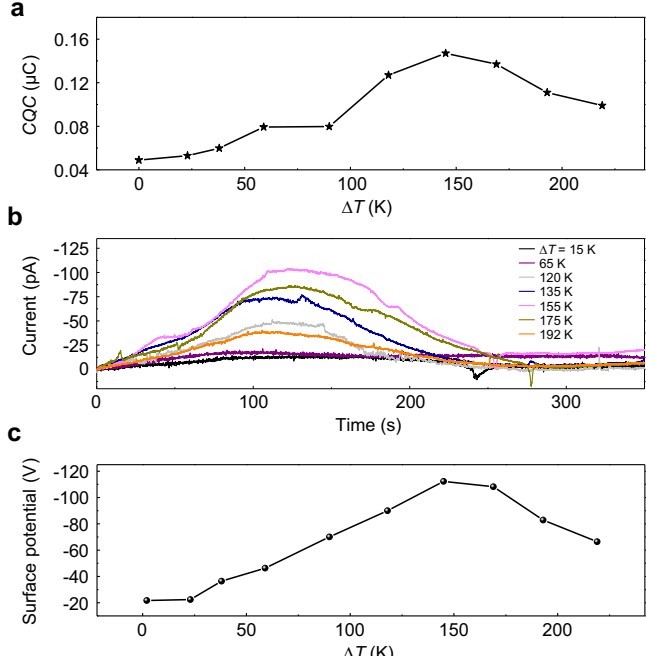

**Fig. 3 Characterization of the effect of $\Delta T$ on TDNG. a** The relationship between transferred charge quantities per cycle ($CQC$) of TDNG at short-circuit condition and $\Delta T$. **b** The thermally stimulated discharge current of Kapton after friction at different $\Delta T$. **c** The surface potential of Kapton after friction at different $\Delta T$.

a current peak is formed. Fig. 3b illustrates the TSD curves of Kapton after friction at different $\Delta T$. When $\Delta T$ increases from 0 K to 192 K, the time integral of current increases from 0 K to 155 K, and decreases when $\Delta T$ is above 155 K, which means the accumulated charges in Kapton reach a maximum at 155 K. Moreover, the surface potentials of Kapton can intuitively reflect the advantages of $\Delta T$ in triboelectrification. As shown in Fig. 3c, the largest surface potential of −112 V has been obtained with $\Delta T$ equals to 145 K and the same trend as mentioned above has been observed. With a fitting analysis, the obtained material-related correction factor $a$ equals to 0.1253, $b$ equals to 165.2682, $C_1$ equals to 0.0248, and $C_2$ equals to 4.256 (Supplementary Fig. 13), which are basically the same as the values obtained from $CQC$. With the certificate of transferred short-circuit current charges, TSD current curves, and surface potential of Kapton, there exists an optimal $\Delta T$ that can effectively boost the output of TDNG through improving the accumulated friction charges in cooler friction layer, and also reveals that the temperature difference influences TDNG's output performance by influencing the transferred charge and friction charge dissipation directly.

**Output performance enhancement of TDNG and its applications**. In addition, the advantages of temperature difference in TENG are illustrated by the dependence of TDNG's output performance on external resistance. As shown in Fig. 4a, the voltage and current show opposite trend with the increased resistance of the external load, and both of them at optimal $\Delta T$ (~145 K) are higher than that without $\Delta T$. The output power of the TDNG reaches a maximum when the external resistance is 3 MΩ. Under the optimal $\Delta T$, the maximum output power increases from 42.2 μW to 206.7 μW (Fig. 4b). Therefore, by combining temperature difference and triboelectrification in TDNG, the power supply capacity of TDNG can be improved effectively (4.9 times of that without temperature difference).

In order to investigate the effectiveness and versatility of the temperature difference effect in TENG and further enhancing output performance of TENG, TDNGs with different friction layers including Al-Kapton, polyamide-6 (PA-6)-Kapton, copper (Cu)-Kapton, iron (Fe)-Kapton, and Al-polytetrafluoroethylene (PTFE) (nanostructure of PTFE and PA-6 are shown in Supplementary Fig. 14) have been studied. As shown in Supplementary Fig. 15 and Fig. 4c, d, the output of all these TDNGs can be improved when $\Delta T$ is constructed between two friction layers. Corresponding to Al-Kapton, PA-6-Kapton, Cu-Kapton, Fe-Kapton, and Al-PTFE TDNGs, the open-circuit voltages have enhanced 2.7, 3.2, 3.0, 3.9, 2.7 times at optimal $\Delta T$, and the short-circuit currents have enhanced 2.2, 2.0, 2.6, 10, 1.7 times at optimal $\Delta T$ (Fig. 4d and Table 1, $V_{opt}$ and $I_{opt}$ are the open-circuit voltage and short-circuit current of the TDNGs at the optimal $\Delta T$, respectively. $V_0$ and $I_0$ are the open-circuit voltage and short-circuit current of the TDNGs when $\Delta T$ equals 0 K, respectively). In this way, the output performance of TDNG can be enhanced effectively by optimizing the friction materials of the TDNG. The optimal $\Delta T$ is related to the materials of the cooler friction layer (as shown in Table 1), where the optimal $\Delta T$ is about 144 K when the cooler friction layer is Kapton. However, when PTFE is used as the cooler friction layer, the optimal $\Delta T$ is just 90 K. Besides, the influence of the strain of friction layers, driving frequency, and contact-separation velocity on output performance have also been studied theoretically and experimentally, respectively. The deformation and displacement of TDNG obtained by the numerical simulation are very small, whose influence can be ignored in contrast to the influence brought by temperature difference (Supplementary Fig. 16). With the driving frequency increases from 0.18 Hz to 0.53 Hz, the voltage increases from 647 V to 770 V (Supplementary Fig. 17), and with the contact-separation velocity increases from 30 mm s$^{-1}$ to 350 mm s$^{-1}$, the voltage increases from 35 V to 753 V (Supplementary Fig. 18). Furthermore, with the help of materials and parameters optimizing, the average peak open-circuit voltage can reach 1.8 kV (Fig. 4e), and an average peak short-circuit current density can reach 480.3 μA cm$^{-2}$ (Fig. 4f) for Al-PTFE TDNG (driving frequency 0.8 Hz, average applied pressure 94.5 kPa, average contact-separation velocity ~1286 mm s$^{-1}$, Supplementary Movie 2). The statistics of ten tested voltage and current results (110 cycles) show that the average open-circuit voltage is 1697±91.1 V, and the average short-circuit current is 443±46.6 μA, corresponding to an average current density of 443±46.6 μA cm$^{-2}$, which is 26.6% larger than the record value (350 μA cm$^{-2}$)[19]. In this way, the effectiveness and universalism of the temperature difference effect in TENG has been investigated, and the highest output current density in contact-separation TENG is achieved simply without complex structure optimization, material preparation or excessively artificial treatment.

Compared with other TENGs, TDNG can be used for harvesting mechanical energy on high-temperature objects more efficiently. Here, a wind-driven TDNG is designed and put on the surface of hot objects to simulate a practical application on high-temperature objects like a car hood, hot road, and roof. As Fig. 5a shows, an etched and fixed Al foil is flattened against the thermostat heater serving as a hotter part of wind-driven TDNG, and an etched 25-μm-thick Kapton flag is swaying in the air serving as a cooler part. The optical image in Fig. 5b demonstrates a working wind-driven TDNG under the wind, and the working behavior, working mechanism, and simulated potential distribution of this wind-driven TDNG are shown in Supplementary Movie 3 and Supplementary Fig. 19. Under continuous 5 m s$^{-1}$ wind, Fig. 5d illustrates the open-circuit voltage and short-circuit current of the wind-driven TDNG in different $\Delta T$. The maximum current can reach 123.3 μA when $\Delta T$ equals to 38.7 K, corresponding to a transferred charge of 185.7 μC. The largest

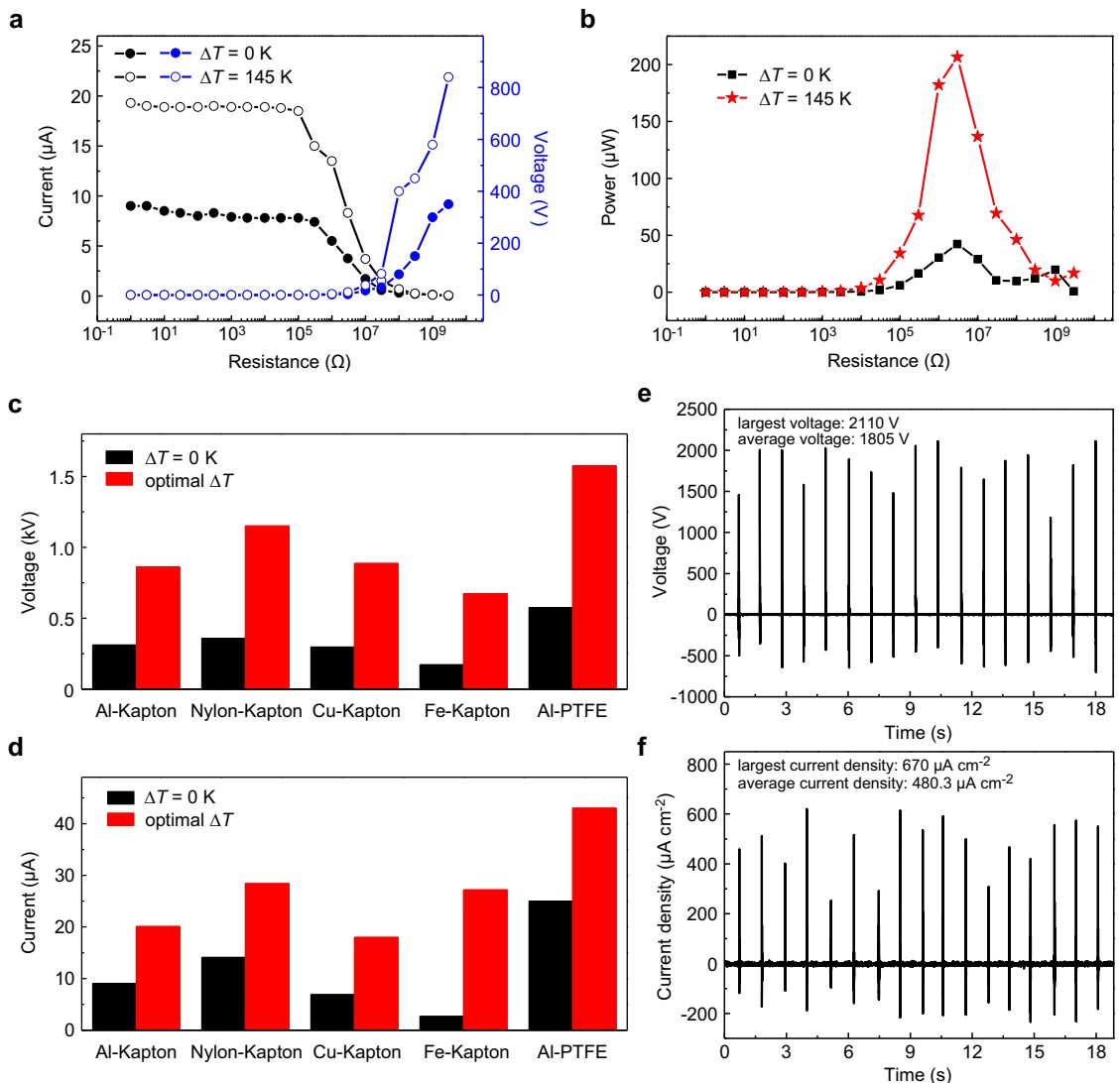

**Fig. 4 The influence of ΔT on TDNG with different friction materials. a** The output voltage (blue line), current (black line) of TDNG with different external circuit load resistance when ΔT is equal to 0 K and 145 K, respectively. **b** The output power of TDNG with different external circuit load resistance when ΔT is equal to 0 K (black line) and 145 K (red line), respectively. **c** The open-circuit voltage and **d** short-circuit current with different friction materials when ΔT = 0 K (black column) and optimal ΔT (red column). **e** The open-circuit voltage and **f** short-circuit current density of Al-PTFE TDNG at ΔT = 90 K.

**Table 1 The ratio $V_{opt}/V_0$ and the ratio $I_{opt}/I_0$ of TDNG composed of different materials with different optimum ΔT.**

| Materials | Al-Kapton | PA-6-Kapton | Cu-Kapton | Fe-Kapton | Al-PTFE |
|---|---|---|---|---|---|
| Optimum ΔT | 145 K | 144 K | 145 K | 143 K | 90 K |
| $V_{opt}/V_0$ | 2.7 | 3.2 | 3.0 | 3.9 | 2.7 |
| $I_{opt}/I_0$ | 2.2 | 2.0 | 2.6 | 10 | 1.7 |

($V_{opt}$ and $I_{opt}$ are the open-circuit voltage and short-circuit current of the TDNGs at the optimal ΔT, respectively. $V_0$ and $I_0$ are the open-circuit voltage and short-circuit current of the TDNGs when ΔT equals 0 K, respectively.)

short-circuit current of 123.3 μA in our work is larger than most previous works (Supplementary Table 1). To effectively demonstrate the TDNG as a high-powered power source, LEDs are directly connected with wind-driven TDNG. As Fig. 5c and Supplementary Movie 4 show, when the temperature difference equals to ~40 K, the wind-driven TDNG can light up 955 white LEDs simultaneously under a wind speed of ~8.3 m s$^{-1}$.

Besides, via a rectifier, a 22 μF capacitor can be charged from 0 V to 7.5 V within 60 s (green line, ΔT equals to 41 K), compared with ΔT equals to 0 K (brown line) and 21 K (yellow line), the capacitor has a faster charging rate, and a higher saturation voltage (Fig. 5e). With the help of a filter circuit (Supplementary Fig. 20), the pulse output is converted to a constant voltage output to drive LEDs continuously. As shown in Fig. 5f and Supplementary Movie 5, when the temperature difference changed from 0 K to 41 K, the red LED, blue LED and white LED are lighted sequentially and continuously, where the light color and brightness can be used as an indicator for temperature difference monitoring. Additionally, after charging for 73 s, a temperature-humidity sensor can be powered by the wind-driven TDNG under conditions of ΔT equals to 102 K when a wind speed is 8.3 m s$^{-1}$ (Fig. 5g, Supplementary Fig. 21, and Supplementary Movie 6).

## Discussion

In summary, the effect of temperature difference on TENG's performance has been investigated through a simulation combining electron-cloud-potential-well model for triboelectrification and the

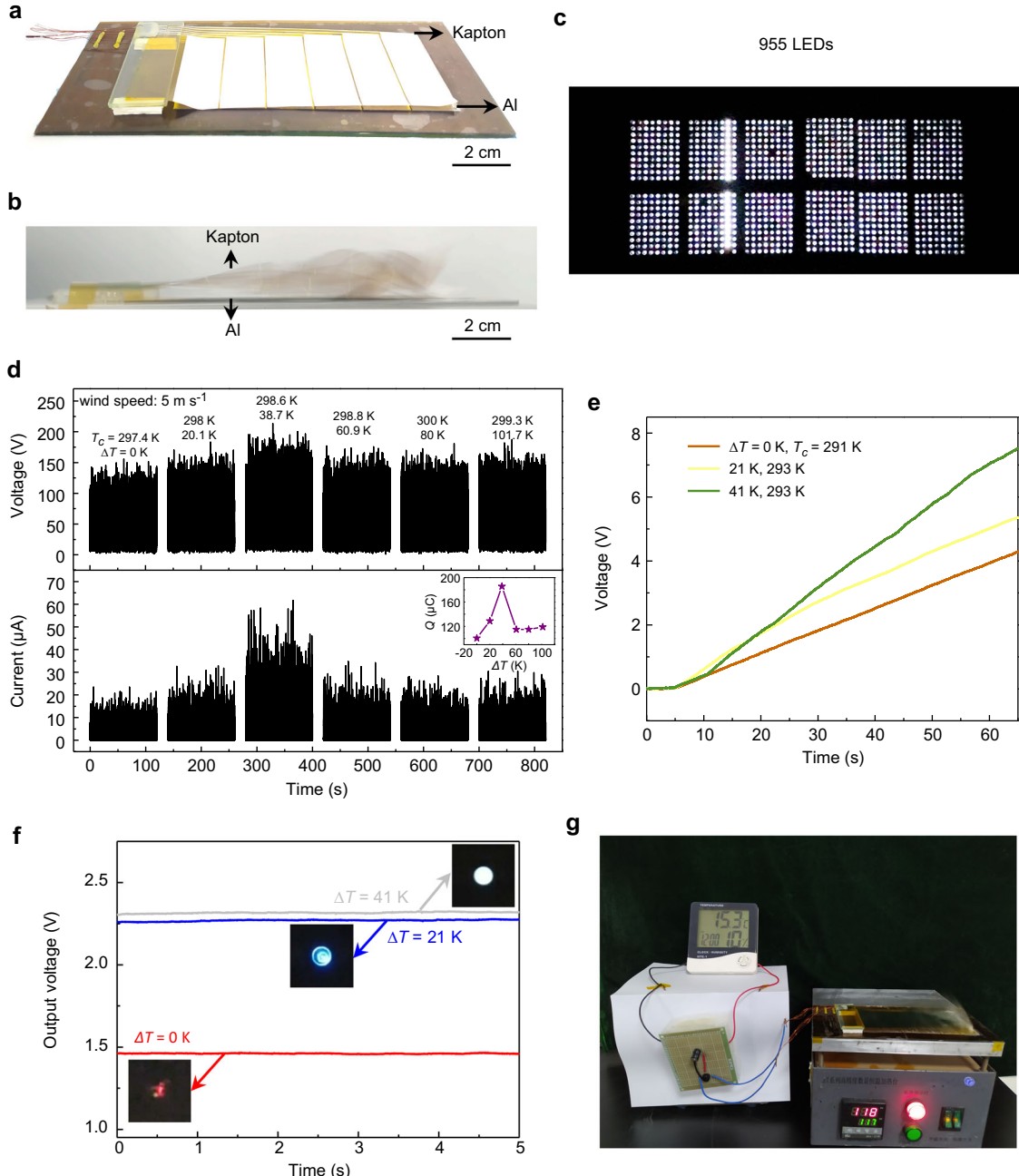

**Fig. 5 Demonstration of the wind-driven TDNG working on the surface of high temperature object. a** The optical image of the wind-driven TDNG. **b** The optical image of a working wind-driven TDNG under the wind. The Al foil is flattened against the thermostat heater serving as a hotter part, and the Kapton flag is swaying in the air serving as a cooler part. **c** 955 white LEDs are lightened by a wind-driven TDNG under a wind speed of ~8.3 m s⁻¹. **d** The open-circuit voltage and the short-circuit current of the wind-driven TDNG under different $\Delta T$ (wind speed: 5 m s⁻¹). **e** Charging curves of a capacitor with 22 μF capacitance by wind-driven TDNG under different $\Delta T$ (0 K (brown line), 21 K (yellow line), 41 K (green line)). **f** LEDs are used for temperature difference indicator. **g** The optical image of a temperature-humidity sensor powered by wind-driven TDNG.

thermionic-emission model, and a type of TENG with controllable friction layer temperature has been designed and fabricated to boost the electrical output performance. With the increase of temperature difference, the output of TDNG increases at first and then decreases due to a tradeoff between electrons transfer from hotter friction layer to cooler friction layer and thermionic emission induced electrons discharge from cooler friction layer. Under the optimal $\Delta T$, the open-circuit voltage, short-circuit current, surface charge density and output power of the Al-Kapton TDNG increase 2.7, 2.2, 3.0 and 4.9 times compared to the case when $\Delta T$ equals to 0 K. The construction of the frication layer temperature difference can be

extended to other TENGs to also boost their outputs. Changing the friction materials from Al-Kapton to Al-PTFE, the current density of TDNG is further enhanced to 443 μA cm⁻², which is 1.26 times the record value (350 μA cm⁻²). Finally, a wind-driven TDNG is demonstrated to power 955 LEDs and a temperature-humidity sensor to show its promising application in an environment with temperature difference.

## Methods

**Fabrication of Al-Kapton TDNG.** Firstly, a piece of Kapton film with a thickness of 100 μm is cut into 5 cm × 5 cm, and ultrasonically cleaned with acetone, ethanol

and deionized water for 15 minutes in sequence. After being dried by nitrogen, Cr/Ag electrode with a size of 4.8 cm × 4.8 cm is sputtered on the middle the Kapton film on one side. The other side of Kapton film is etched by RIE for 1 h with 3 sccm Ar and 7 sccm CF$_4$, 250 W input power. Secondly, a piece of cleaning Al film with a thickness of 20 μm are cut into 5 cm × 5 cm and then immersed in 0.5 mol L$^{-1}$ NaOH solution for 2 mins to fabricate Al nanostructures. After being washed by deionized water, the etched Al film is dried in a 333 K oven. Finally, placing the Al foil on a thermostat heater to form a hotter part and Kapton film on a water-cooling system to form a cooler part. The hotter part and cooler part compose the TDNG and air gap between the hotter part and cooler part is 5 cm.

**Fabrication of the wind-driven TDNG.** The wind-driven TDNG is designed by replacing the 5 cm × 5 cm, 100-μm-thick Kapton film and the 5 cm × 5 cm Al foil of the Al-Kapton TDNG with the 15 cm × 7 cm, 25-μm-thick Kapton film and the 15 cm × 7 cm Al foil as friction layers, while keeping all other materials and processes unchanged. Different from the Al-Kapton TDNG, the Al film is placed on a thermostat heater to form a hotter part and the Kapton film is hanging in wind to form a cooler part. The air gap between the hotter part and cooler part is 1 cm.

**Measurement and characterization.** Morphology of the samples are characterized by emission scanning electron microscopy Apreo S. The output current signals of TDNG are measured by a low-noise current preamplifier Stanford Research SR570. For the measurements of the wind-driven TDNG, a commercial air gun is used and a hand-held anemometer is used for measuring wind speed.

## Data availability
The data that support the findings of this study are available from the corresponding author upon reasonable request.

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

## Acknowledgements
We sincerely acknowledge the support from Joint fund of Equipment pre-Research and Ministry of Education (No. 6141A02022518), the National Program for Support of Top-notch Young Professionals. We also thank Dr. Gaoda Li, Weihao Gao, Binbin Ma, Chang Lu, Xiaoyu Huang for help with the fabrication of devices and the construction of experimental equipment. The authors also thank Dr. Qiong Deng for the calculation discussion.

## Author contributions
B.L.C., Y.Q.D. and Q.X. contributed equally to this work. Y.Q., B.L.C. and Y.Q.D. designed the device, B.L.C., Q.X. and S.B. fabricated and measured the device, X.F.J. and Q.X. conducted the simulation via COMSOL, B.L.C., Y.Q.D., S.B., J.W. and Y.Y.D.C. analyzed the experimental data, plotted the figures and prepared the manuscript, and all authors reviewed and commented on the manuscript.

## Competing interests
The authors declare no competing interests.
