## [Peer Review File · Nature Communications]

REVIEWER COMMENTS

Reviewer #1 (Remarks to the Author):

This work reported a triboelectric nanogenerator utilizing the temperature difference to promote charge transfer. And performance under different temperature differences and some different friction materials are considered. However, in this work needs to be improved for the high-level journal of Nature Communications. Some reasons are given below,

As mentioned, the largest short-circuit current density reaches $396 \mu\text{A}/\text{cm}^{-2}$ as shown in Figure. 4f. The question is that why the peak currents are so uneven unlike those in Figure 2b. More importantly, considering the area of device in this work is 25 cm^2 , so the calculated current is 9.9 mA ? This data should be further proved to convince us.

The output charge when the largest short-circuit current density obtained should be given. At the characterization part, SR570 is used to measure current, and how do you measure current which is out the measurement range, such as $396 \mu\text{A}/\text{cm}^{-2}$. Besides, the voltage measurement for up to thousands of volt should be stated clearly.

This paper focuses the impact of temperature difference on triboelectric nanogenerator. What is different between your work and other works about the studying of the influence of temperature changing to triboelectric nanogenerator?

Only the temperature difference is considered. Is there any relationship between the temperature of the hot part and cool part to the performance of triboelectric nanogenerator? That is, is the temperature difference set at one constant high/low temperature while different low/high temperatures? And would the performance be unchanged at the same temperature difference while different respective high and low temperature?

Performance is closely related to the working frequency, however, working frequency is omitted in this work. It is obviously that the frequency in Figure 4e, f is much higher than that in Figure. 2b, and so the corresponding current is much higher. Besides, to compare the performance with other works, the related parameters should be the same, especially the working frequency when the current is compared.

The triboelectric nanogenerator contains a cooler part and a hotter part, but how to cool and heat the part is not clear.

At line 232, it is mentioned that "after a short-time storage", this sentence is so vague! What is the exact short the storage time? It is required to plot the specific charging curve and mark the storage time. The related video should be given.

As authors mentioned, the current is high in this work, so this triboelectric nanogenerator should be able to power many LEDs. However, only two LEDs with not bright light are showed, demonstrating a quite weak of the ability in actual application of this triboelectric nanogenerator. And in generally, the application video should be demonstrated.

Reviewer #2 (Remarks to the Author):

Inspired by the detailed experimental and modeling results in reference [24] (Advanced Materials, 31, 1808197, (2019)), the authors present a high-performance contact-separation temperature difference TENG with (TDNG) controllable friction layer temperature (TDNG) in this paper. The modeling and experimental results show that the device output can be enhanced by optimizing the temperature difference between the hotter and cooler friction layers of the TDNG device. It is noted that the output current density of one such device is $396 \mu\text{A}/\text{cm}^2$ when the temperature difference is 145 K . This is a 13% higher than the previous record of $350 \mu\text{A}/\text{cm}^2$. Similar temperature effects on device performance are also observed in TDNG with different friction layers such as Al-Kapton, PA6-Kapton, Cu-Kapton, Fe-Kapton and Al-PTFE. A wind-drive TDNG is demonstrated to power a temperature-humidity sensor, verifying the potential applications of such devices. Some important information is missing in the current version of the manuscript.

- (1) Surface charge density is the key factor for high-performance triboelectric nanogenerators. Have the authors measured the surface charge density at different temperature differences? Is the surface charge density enhanced due to the change of temperature difference? Is the enhanced output current mainly attributed to the increase in surface charge density or other factors?
- (2) In Equation (2): What is "A"? What are the material parameter values used in calculating the curve in Figure 1(b)?
- (3) What is the purpose of the Al film treatment including the immersing in 0.5 mol/L NaOH solution for 2 minutes and being washed by deionized water?
- (4) The Al film is placed on a hot plate and used as the hotter side of TDNG. What is the temperature (or the temperature range) of the hot plate? If the temperature of the Al film is raised 219 degree C above that of the cooler side (kept at 299 K), it is very likely that a thin oxide layer (Al₂O₃) is quickly formed on the surface of the Al film. This Al₂O₃ may serve as an insulating layer that could prevent charge transfer between the hotter side and the cooler side of the device. The proposed schematic in Figure 1 (a) ignores the existence and the possible role of this insulating layer, why?
- (5) The Al film temperature is assumed to be the same as the setting temperature of the hot plate, which is reasonable. The cooler side of the device physical touches the Al film in one contact-separation cycle. How to monitoring the temperature of the cooler side of the device? How to ensure the temperature of the cooler side always remains at 299 K regardless of the rising temperature of the hotter side? Can the authors provide a photo (or a short video) of the experimental setup and show a real device is under test?
- (6) Can the author provide more details about their COMSOL simulation (for Figure 1(b) and Figure 1(c))? Such as material parameters, specific model that was used, device dimensions, etc.
- (7) How was the 396 $\mu\text{A}/\text{cm}^2$ obtained from Figure 4 (f)? Is it the peak value, or peak-to-peak value? Or is it the average value? What is the size of the Al-PTFE TENG device that was used for this test? What is the actually measured short-circuit current?
- (8) Figure 5 (a) does not show a clear picture of the application test. Can the author upload a short video of the wind-driven TDNG device in operation?
- (9) The device performance is maximized when the temperature difference between the hotter and cooler friction layers is above 100 degree C. Can the authors suggest/propose a few practical scenarios in which the temperature difference can reliably be maintained at above 100 degree C?

Reviewer #3 (Remarks to the Author):

The authors attempted to optimize the temperature difference between friction layers for high output of triboelectric generators (TENGs). This ideal is novel and interesting based on simulation combining the electron-cloud-potential-well model for triboelectrification and the thermionic-emission model. However, the mechanistic analysis and the experiments of the TENG device were not carried out systematically and carefully. For example, the relationships among the working temperature, the temperature difference, and the outputs of TENGs have not been discussed adequately. Moreover, in practice, the influence of the heat conduction between the friction layers on the output of TENGs has not been addressed carefully. Several key questions are left without clear answers, such as the rational of inducing temperature difference to increase the output, the influence of heat conduction on the output of the TENG, and how to maintain the temperature difference. Therefore, this manuscript is not suggested to be published in its present form. More specific comments are listed below.

1. On page 4, the authors stated that "On the one hand, during friction, more electrons will hop from the hotter friction layer to cooler friction layer (middle part of Figure 1a) to enhance the charge density. On the other hand, the increased temperature of the cooler friction layer will discharge the electrons accumulated in the cooler friction layer (electrons are easier to escape out of the potential well, and get back to the hotter friction layer in contact processes or spill into air, as the right part of Figure 1a shows), and lead to reduce charge density and output performance

of TENG." This statement is suggested to provide some evidence.

2. On Page 5, how to obtain the equation (1) and (2)? What are the basic assumptions for these equations? And what is the relationship between equation (1) and (2)? Moreover, these equations may have spelling mistakes, where ΔT is not included?

3. On page 8, the authors stated that "Furthermore, with the help of materials optimizing, the largest open-circuit voltage can reach to 1.5 kV (Figure 4e), and a largest short-circuit current density can reach to 396 $\mu\text{A}/\text{cm}^2$ (Figure 4f) for Al-PTFE TDNG, which is 13% larger than the record value (350 $\mu\text{A}/\text{cm}^2$).19". The repeatability of the peak outputs of voltage and current is normally weak. The authors utilized the largest peak current density to compare with the output of the reference. It seems that this comparison is unfair as well as the improvement is very slight. Because the variation of these measurements should be considered carefully. More importantly, such peak current density relies on the separation speed of the friction layers.

4. As shown in Figures 2 and 3, the surface profile of the friction layer has not been quantitatively described. Only small surfaces were illustrated. And the temperature changes were up to 219 K. The mechanical properties of these friction layers had a huge change during the measurements. As well as the real contact area changed. However, the influence of the change of the real contact area on the output was not discussed in the manuscript. The authors are suggested to consider this phenomenon carefully.

5. The size of the demonstration in Figure 5a should be given. The wind speed of 15 m/s is very large. More demonstrations based on smaller wind speed as common cases are suggested. Moreover, such an output is not surprising. Some benchmarkers from other references are suggested to be added for comparison.

Dear Dr. Ronghuan Zhang and reviewers,

Thank you very much for the helpful comments and suggestions. We have carefully revised the manuscript and supporting information in accordance with the comments of editor and reviewers. The itemized responses to the comments are included under below. The revised manuscript and supporting information are uploaded. A copy of revised manuscript and Supplementary Information with highlighted changes is also uploaded for your convenience.

With best regards,

Yours sincerely,

Yong Qin

Professor, Lanzhou University

Response to the reviewers' comments

Response to reviewer 1

Comments to the Author

This work reported a triboelectric nanogenerator utilizing the temperature difference to promote charge transfer. And performance under different temperature differences and some different friction materials are considered. However, in this work needs to be improved for the high-level journal of Nature Communications. Some reasons are given below.

Response: We appreciate very much the reviewer's very positive and valuable comments. And we have carefully revised our manuscript according to these important comments.

1. As mentioned, the largest short-circuit current density reaches $396 \mu\text{A}/\text{cm}^2$ as shown in Figure. 4f. The question is that why the peak currents are so uneven unlike those in Figure 2b. More importantly, considering the area of device in this work is 25 cm^2 , so the calculated current is 9.9 mA ? This data should be further proved to convince us.

Response: We appreciate this comment from the reviewer. In Figure 2b, the TDNG is driven by a computer-controlled linear motor and the even output of TDNG is owing to the regular movement of the motor (as shown in **Supplementary Movie 1**). Whereas, in Figure 4e-f, the TDNG is driven by irregular human mechanical motion (patting, **Supplementary Movie 2**), so the output signal is not even, correspondingly.

The area of the device in Figure 4e-f is 1 cm^2 , so the corresponding largest short-circuit current is $396 \text{ }\mu\text{A}$ (**Fig. R1a**) rather than 9.9 mA . We have stated this in the revised manuscript to make it clearer. Besides, according to your sixth comment, the data in Figure 4f are retested using similar parameters (driving frequency 0.8 Hz , applied pressure 94.5 kPa) to the reference (driving frequency $\sim 0.8 \text{ Hz}$, applied pressure $\sim 150 \text{ kPa}$; *ACS Nano* 2019; 13, 4640-4646), and the figures in the manuscript are replaced, correspondingly. Due to the limitation of the equipment, the 150 kPa pressure can't be applied. The average applied pressure is 94.5 kPa in our work. At the optimal temperature difference of 90 K , the largest short-circuit current of the PTFE-Al TDNG can reach $670 \text{ }\mu\text{A}$, and the average current is $480.3 \text{ }\mu\text{A}$ corresponding to an average current density of $480.3 \text{ }\mu\text{A cm}^{-2}$. After the statistical analysis of ten tested voltage and current results, the average short-circuit current is $443 \pm 46.6 \text{ }\mu\text{A}$, corresponding to an average short-circuit current density of $443 \pm 46.6 \text{ }\mu\text{A cm}^{-2}$. Although the average applied pressure is only 63% of that in reference (150 kPa), the short-circuit current density is 26.6% larger than that of previous work ($350 \text{ }\mu\text{A cm}^{-2}$, *ACS Nano* 2019; 13, 4640-4646). This important information has been added to the manuscript and the Supplementary Information.

Fig. R1 **a** The short-circuit current corresponding to Figure 4f. **b** The retested short-circuit current of TDNG with similar parameters to the reference.

2. The output charge when the largest short-circuit current density obtained should be given.

Response: Thanks for your comment. The average output charge is 128 nC in the original manuscript, and the average output charge is 150 nC after the testing parameters are tuned to be comparable with the reference (*ACS Nano* 2019; 13, 4640-4646) in the revised manuscript according to your sixth comment. The output charge quantity (Q) can be obtained by integrating the largest current pulse with time. In the last manuscript, the output charge quantity in a single contact process is 171 nC, and in a single separation process is 85 nC, corresponding to an average output charge quantity of 128 nC (**Fig. R2b**). After the testing parameters are tuned to be comparable with the reference (*ACS Nano* 2019; 13, 4640-4646), the output charge quantity can reach 163 nC in a single contact process and -119 nC in a single separation process, respectively, corresponding to an average output charge of 141 nC (**Fig. R2d**). Furthermore, as shown in **Table R1**, after the statistical analysis of 17 cycles of current data, an average output charge quantity of 150 ± 17.8 nC is obtained. This important information has been added to the manuscript and the Supplementary Information.

Fig. R2 The output current and charge of TDNG. **a** The short-circuit current output corresponding to Figure 4f. **b** The output charge quantity of TDNG corresponding to Figure 4f. **c** The retested short-circuit current of TDNG with similar parameters to the reference. **d** The output charge quantity of TDNG.

Table R1 The output charge quantity of TDNG obtained by the time integral of 17 cycles of current data.

number of cycles	1	2	3	4	5	6	7	8	9
Q (nC)	163	141	159	156	116	155	126	173	156
number of cycles	10	11	12	13	14	15	16	17	-
Q (nC)	144	156	131	149	124	159	180	167	-

3. At the characterization part, SR570 is used to measure current, and how do you measure current which is out the measurement range, such as 396 $\mu\text{A}/\text{cm}^2$. Besides, the voltage measurement for up to thousands of volt should be stated clearly.

Response: Thank you very much for your comment. We are very sorry for not describing the measurement clearly. The area of the device in Figure 4e-f is 1 cm^2 , so the corresponding largest short-circuit current is 396 μA (**Fig. R3a**) rather than 9.9 mA, the 396 μA current is in the measuring range of SR570. We have stated this in the revised manuscript to make it clearer. Besides, according to your sixth comment, after the testing parameters is tuned to be comparable with the reference (*ACS Nano* 2019; 13, 4640-4646), the corresponding largest peak current 670 μA (**Fig. R3b**) is also in the measuring range of SR570.

The voltage of TDNG is measured by SR560 and a voltage divider. We are sorry that we didn't explain this in the last manuscript, we have added this information in the revised Supplementary Information. The largest measuring range of SR560 is ± 3 V. In order to measure the output voltage of TDNG up to thousands of volts, a voltage divider (the voltage divider is a passive linear circuit that produces an output voltage that is a fraction of its input voltage) is used. As shown in **Fig. R3c**, the voltage divider is composed of ten 100-M Ω -resistors, nine 10-G Ω -resistors, and nine 1-G Ω -resistors, whose whole resistance is 100 G Ω . The total resistance of 100 G Ω is designed to match the internal resistance of TDNG. When the external resistance is greater than 30 G Ω , the

end voltage is consistent with the open-circuit voltage of TDNG. The 100-M Ω -resistor (R_1) in voltage divider is used to match the input impedance of SR560 (100 M Ω). As shown in **Fig. R3c**, when the output voltage of TDNG (U_{input}) is applied across the series resistance and the output is the voltage across R_1 . Since the current (I) flowing through each resistor in the series circuit is equal, the relationship between the input voltage (U_{input}) and the output voltage (U_{output}) is:

$$U_{output} = \frac{R_1}{R_w} U_{input} \quad (R1)$$

where R_w is the whole resistance of series resistances. Because the whole resistance of series resistances is 100 G Ω , R_1 is 100 M Ω , and the output voltage of TDNG (U_{input}) can be obtained as follows

$$U_{input} = 1000 \times U_{output} \quad (R2)$$

In this way, the measurement range of SR560 can be enhanced to ± 3000 V, and the output voltage of TDNG up to thousands of volts can be measured.

Fig. R3 The short-circuit current of TDNG and the schematic diagram of the voltage divider. **a** The short-circuit current corresponding to Figure 4f. **b** The retested short-circuit current with similar parameters to the reference. **c** The schematic diagram of the voltage divider.

4. This paper focuses the impact of temperature difference on triboelectric nanogenerator. What is different between your work and other works about the studying of the influence of temperature changing to triboelectric nanogenerator?

Response: We appreciate the reviewer for this comment. Previous work mainly focused on the influence of temperature on the output performance of TENGs and the mechanism of temperature influence on TENGs, such as the thermionic emission of electrons, the atomic thermal vibration, surface state, *etc.* There are three differences between our work and previous works. Firstly, comparing with the preliminary principle study with the assistance of Atomic Force Microscopy instrument in previous works, we developed an actual TENG utilizing the temperature differences to harvest both mechanical energy and heat energy in the environment. Secondly, the influences of high temperature and low temperature on the output of TENG are studied, and an optimal temperature difference is found in our work. Thirdly, under the optimal temperature difference, the output of our TDNG is higher than the reported highest output performance.

5. Only the temperature difference is considered. Is there any relationship between the temperature of the hot part and cool part to the performance of triboelectric nanogenerator? That is, is the temperature difference set at one constant high/low temperature while different low/high temperatures? And would the performance be unchanged at the same temperature difference while different respective high and low temperature?

Response: We appreciate the reviewer for the very nice comments. Indeed, the performance of the triboelectric nanogenerator is also related to the temperature of the

hot part and cool part, and will change at the same temperature difference while different respective high and low temperature according to the following theoretical analysis.

The transferred charge is decided by the temperature of the hotter part (T_h) and cooler part (T_c). For the hotter part, the transferred charge during the friction can be described as an empirical law (*Adv. Mater.* 2019; 31, e1808197):

$$\sigma = -C_1 T_h + C_2 \quad (\text{R3})$$

where σ is the transferred charge density when the temperature of the cooler friction layer keeps fixed, T_h is the temperature of the hotter friction layer, C_1 and C_2 are the material-related correction factors. As the temperature of the hotter part increases, the transferred charge density σ will increase, which can enhance the surface charge density of the cooler part (σ_c). On the other hand, the temperature of the cooler part (T_c) will rise with T_h increases due to the heat exchange in contact-separation processes and air heat transfer. The surface charge density of the cooler part follows an exponential decay during thermionic emission, which can be described as follows (*Adv. Mater.* 2018; 30, e1706790)

$$\sigma_c = e^{-SA t} \sigma_{c0} \quad (\text{R4})$$

where σ_c is the surface charge density, and σ_{c0} is the initial value of σ_c , A is the surface area of TDNG, and t is the time of heat balance between the cooler part and hotter part, $S = \frac{\lambda_1 A_0}{k} T_c e^{\frac{q\psi}{kT_c}}$, where λ_1 is the material-specific correction factor, A_0 is Richardson constant of a free electron, k is Boltzmann constant, and T_c is the temperature of the cooler friction layer. Normally the increase of T_h will increase the transferred charge density, which is beneficial to the charge accumulation of the cooler part. At the same time, the increase of T_c will increase the charge loss due to the thermionic emission. These two factors compete with each other, resulting in a complicated trend of the surface charge density of the cooler part. Based on mentioned above, combining equation (R3) and equation (R4), the surface charge density of TDNG can be described as follows

$$\sigma_{TDNG} = (-C_1 T_h + C_2) e^{-SA t} \quad (\text{R5})$$

In our experiments and simulations, due to the heat exchange in contact-separation processes and air heat transfer, T_c increases approximately linearly with T_h , which can be approximately described as $T_c = aT_h + b$, where a and b are material-related correction

factors, which is related to the thermal conductivity of the materials of the cooler part and the heat transfer efficiency. In this way, T_c and T_h can be unified by temperature difference ($\Delta T = T_h - T_c$), as $(a\Delta T+b)/(1-a)$ and $(\Delta T+b)/(1-a)$. Substituting T_c and T_h into equation (R5), we can obtain the relationship between the surface charge density of TDNG and temperature difference, which can be described as follows

$$\sigma_{TDNG} = (-C_1 \frac{\Delta T+b}{1-a} + C_2) e^{-Sat} \quad (R6)$$

where $S = \frac{\lambda_1 A_0 (a\Delta T+b)}{k(1-a)} e^{\frac{qv(1-a)}{k(a\Delta T+b)}}$. Therefore, under the competition of the electron thermal emission of the hotter part and the cooler part, the surface charge density of TDNG will increase first then decrease when ΔT increases. Combining the relationship between surface charge density and output, the relationship between the output performance of TDNG (V) and the temperature difference can be written as follows

$$V = \frac{Q}{C} = \frac{4\pi k_e \sigma_0}{\varepsilon \varepsilon_r} x(t) = -\frac{4\pi k_e x(t)}{\varepsilon \varepsilon_r} (-C_1 \frac{\Delta T+b}{1-a} + C_2) e^{-Sat} \quad (R7)$$

where k_e is Coulomb's constant, $x(t)$ is the distance between two friction layers (which is a function of time), ε is the relative permittivity of the cooler friction layer, and ε_r is the relative permittivity of vacuum.

According to equation (R7), the output of TDNG is mainly related to the temperature difference between two friction layers; at the same time, it is also related to the material-related correction factors, such as the thermal conductivity and the heat transfer efficiency. Because the relationship between T_c and T_h is a constant function in one heat transfer system, the temperature difference cannot be set as a constant with different T_c and T_h in one heat transfer system. Suppose we want to obtain the output performance of TENG with the same temperature differences but different T_c and T_h . In that case, the heat transfer system must make corresponding changes, such as changing the thermal conductivity of the material. In this way, when the temperature difference is set at one constant high/low temperature while different low/high temperatures, the material-related correction factor a and b will change. As shown in **Fig. R4a**, when the thermal conductivity of the friction layer in the cooler part increases from $0.06 \text{ W m}^{-1}\text{K}^{-1}$ to $0.18 \text{ W m}^{-1}\text{K}^{-1}$, the same temperature differences but different T_c and T_h can be obtained. Responsively, the material-related correction factor a will change from 0.224

to 0.101. As shown in **Fig. R4b**, when the thermal conductivity of the friction layer in the cooler part changed, the surface charge density at different T_h and different T_c but the same ΔT can be obtained. At the same ΔT , the output of TDNG is related to the thermal conductivity of the materials, the larger thermal conductivity is, the higher the optimal temperature difference is, and the larger the surface charge density is. In other words, the output performance of TDNG would be changed at the same temperature difference while different high and low temperatures. Besides, according to your valuable suggestion, we will systematically study it experimentally in the following work.

Fig. R4 The influence of the friction layer's thermal conductivity on the output of TDNG. **a** The influence of the thermal conductivity of the friction layer on the temperature of the cooler part. **b** The influence of the thermal conductivity of the friction layer on the surface charge density of the cooler part.

6. Performance is closely related to the working frequency, however, working frequency is omitted in this work. It is obviously that the frequency in Figure 4e, f is much higher than that in Figure. 2b, and so the corresponding current is much higher. Besides, to compare the performance with other works, the related parameters should be the same, especially the working frequency when the current is compared.

Response: We appreciate the reviewer very much for this comment. The output performance of TDNG is related to the working frequency. The driving frequency is 0.7 Hz in Figure 2b and is ~4.4 Hz in Figure 4e. According to your comment, to make our result is comparable with other works, the data in Figure 4e-f are retested. Besides the working frequency, TENG's performance is also related to the contact-separation velocity as the reviewer #3 pointed out. In the new test, similar parameters (driving frequency 0.8 Hz, applied pressure 94.5 kPa, contact-separation velocity 1286 mm s⁻¹) to the reference (driving frequency ~0.8 Hz, applied pressure ~150 kPa, contact-separation velocity > 1286 mm s⁻¹; *ACS Nano* 2019; 13, 4640-4646) are used. Although the applied pressure and contact-separation velocity are lower than the reference, the short-circuit current density of our work is higher than the previous work. Below are the specific test results.

Due to the limitation of our measurement equipment, the 150 kPa pressure can't be applied. The average applied pressure is 94.5 kPa in our work. At the optimal temperature difference of 90 K, the largest open-circuit voltage is 2110 V, and the average open-circuit voltage is 1805 V (**Fig. R6a**); the largest current can reach 670 μ A, and the average current is 480.3 μ A (**Fig. R6b**) corresponding to an average current density of 480.3 μ A cm⁻² (the area of TDNG is 1 cm²). The statistics of 10 tested voltage and current results show that the average open-circuit voltage is 1697 \pm 91.1 V, and the average short-circuit current is 443 \pm 46.6 μ A, corresponding to an average current density of 443 \pm 46.6 μ A cm⁻². Although the average applied pressure is just 63% of that in reference, the short-circuit current density is 26.6% larger than that of previous work (350 μ A cm⁻², *ACS Nano* 2019; 13, 4640-4646). Besides, the contact-separation velocity in our work is ~1286 mm s⁻¹, and the contact-separation velocity in the compared reference is larger than 1286 mm s⁻¹ (This can be inferred from the data of the amount of charge transferred. With similar output, the amount of transferred charge in the compared reference (20.2 nC cm⁻²) is smaller than our experimental value (150 \pm 17.8 nC cm⁻²), indicating that the width of the current pulse is narrower than our work and the contact-separation velocity is larger than that used in our work). This important information has been added to the manuscript and the Supplementary Information.

Fig. R5 The output of TDNG under similar situations in reference. **a** The open-circuit voltage of TDNG under similar situations in reference. **b** The short-circuit current of TDNG under similar situations in reference.

Furthermore, the influence of the driving frequency and the contact-separation velocity on the output performance have been studied systematically. As shown in **Fig. R6a-b**, when the working frequency increases from 0.179 Hz to 0.526 Hz, the largest open-circuit voltage of TDNG increases from 647 V to 770 V. When the contact-separation velocity (v) increases from 30 mm s⁻¹ to 350 mm s⁻¹, the open-circuit voltage of TDNG increases from 35 V to 753 V (**Fig. R6c-d**).

Fig. R6 The relationship between driving frequency, contact-separation velocity, and performance of TDNG. **a** The voltage of TDNG under different frequencies (from 0.179 Hz to 0.526 Hz). **b** The peak voltage under different frequencies. **c** The voltage of TDNG under different contact-separation velocity (from 30 mm s⁻¹ to 350 mm s⁻¹). **d** The peak voltage under different contact-separation velocity.

7. The triboelectric nanogenerator contains a cooler part and a hotter part, but how to cool and heat the part is not clear.

Response: Thank you very much for your comment. We are sorry that haven't described it clearly. The hotter part is heated by a thermostat heater, and the cooler part is cooled by a water-cooling system. As shown in the schematic diagram in **Fig. R7a** and the picture in **Fig. R7b** shown, in experiments, the hotter part is composed of a thermostat heater and a friction layer, where the thermostat heater can provide a stable temperature for heating the friction layer (ranging from 298 K to 623 K). The cooler part is composed of a water-cooling system, friction layer, and Cr/Ag electrode. The water-cooling system can provide a stable temperature for cooling the friction layer and reduce the heat exchange influence between two friction layers. In Figure 5, the cooler part is cooled by the cold wind, and the thermostat heater heats the hotter part. Furthermore, in real applications, the cooler part can be cooled by natural cold sources, such as the cold wind (~293 K), cold water (~283 K), frozen soil (~273 K), ice (~273 K), *etc.* The hotter part can be heated by natural heat sources and heat sources in life, such as pavements, roofs after exposure (~ 348 K), car engines (358-378 K), exhaust pipes (593-673 K), the surface of high-speed aircraft (505-810 K), *etc.* Under the cooling effect of the cold source, the temperature of the cooling part can be well maintained near the temperature of the cold source. The temperature of the hotter part depends on the environment in which the TDNG is used. For instance, when the cold wind (~293 K) is used as the cold source in the cooler part and the exhaust pipes (393-673 K) is used as the heat source in the hotter part, the temperature difference can be maintained ~100-380 K. When the cold water (~283 K) is used as the cold source in the cooler part and the car engines (358-378 K) is used as the heat source in hotter part, the temperature difference can be maintained as 70-

95 K. This important information has been added to the manuscript and Supplementary information.

Fig. R7 The schematic diagram and picture of TDNG in experiments. **a** The schematic diagram of TDNG in experiments. **b** The optical photograph of TDNG in experiments.

8. At line 232, it is mentioned that “after a short-time storage”, this sentence is so vague! What is the exact short the storage time? It is required to plot the specific charging curve and mark the storage time. The related video should be given.

Response: We appreciate the reviewer very much for the comment. We are very sorry for this vague description. In our work, the average charging time is about 73 s. This sentence has been revised to “after charging for 73 s” in the revised manuscript. The charging and discharging curves of two charging-discharging cycles are shown in **Fig. R8a**, and corresponding photos of discharging processes are shown in **Fig. R8b**. The storage time is marked in two charging-discharging cycles, where the first charging time is 71 s, and the second charging time is 74 s. The average charging time of 73 s is calculated with five charging-discharging cycles. Besides, the corresponding video of the temperature-humidity sensor powered by wind-driven TDNG (**Supplementary Movie 6**) has also been uploaded. This part has been added to the manuscript and the Supplementary Information.

Fig. R8 Powering for the temperature-humidity sensor. **a** The charging curve of the temperature-humidity sensor. **b** The optical photograph of the first discharging process. **c** The optical photograph of the second discharging process.

9. As authors mentioned, the current is high in this work, so this triboelectric nanogenerator should be able to power many LEDs. However, only two LEDs with not bright light are showed, demonstrating a quite weak of the ability in actual application of this triboelectric nanogenerator. And in generally, the application video should be demonstrated.

Response: We appreciate the reviewer very much for the comment. In Figure 5, the output of the TDNG driven by wind is filtered to a constant voltage/current output and then used to driven the LEDs, so only two light LEDs are showed. If the LEDs are directly connected with wind-driven TDNG, 1882 white LEDs can be lightened up simultaneously, which is 682 LEDs more than previous works (*Nano Res.* 2018; 11, 1873-1882). The reason to add a filter circuit between the TDNG and the LEDs, and the experimental results for lighting 1882 LEDs are listed below:

In Figure 5c, the brightness and lighted number of LEDs are used as an indicator for temperature differences monitoring. Since the output of the TDNG is a pulse signal, if it is directly used for powering LEDs, the brightness of LEDs will change with time, which is not applicable for temperature differences monitoring. Therefore, with the help of the filter circuit (**Supplementary Fig. 16**), the pulse output is converted to a constant voltage/current (**Fig. R9a**) output to drive LEDs continuously. Because of the lower

output voltage after filtering (< 2.5 V), only two LEDs with not bright light are shown in the last manuscript.

Besides, we found that using the intensity of the same light color (like blue light) as an indicator of temperature differences is difficult to distinguish. We changed the temperature differences indicator to a parallel circuit composed of a red LED, a blue LED, and a white LED. As shown in **Fig. R9a** and **Supplementary Movie 5**, when the temperature difference changed from 0 K to 38.7 K, the red LED, blue LED, and white LED are lit sequentially and continuously. In this way, the light color and brightness can be used as an indicator for temperature differences monitoring more precisely.

According to the reviewer's comment, to effectively demonstrate the TDNG as a high-powered power source, LEDs are directly connected with wind-driven TDNG. As **Fig. R9b** and **Supplementary Movie 4** shown, when the temperature difference equals to ~ 40 K, the wind-driven TDNG can light up 1882 white LEDs simultaneously, which is 682 LEDs more than previous works (*Nano Res.* 2018; 11, 1873-1882). At the same time, the driven wind speed (~ 8.3 m s⁻¹) used in our work is far lower than that used in previous work (15 m s⁻¹). This important information has been added to the manuscript and the Supplementary Information.

Fig. R9 The application demonstration of the wind-driven TDNG at different temperature difference. **a** Different LEDs are used for temperature difference indicator. **b** 1882 white LEDs are lightened by a wind-driven TDNG.

Response to reviewer 2

Comments:

Inspired by the detailed experimental and modeling results in reference [24] (Advanced Materials, 31, 1808197, (2019)), the authors present a high-performance contact-separation temperature difference TENG with (TDNG) controllable friction layer temperature (TDNG) in this paper. The modeling and experimental results show that the device output can be enhanced by optimizing the temperature difference between the hotter and cooler friction layers of the TDNG device. It is noted that the output current density of one such device is $396 \mu\text{A}/\text{cm}^2$ when the temperature difference is 145 K. This is a 13% higher than the previous record of $350 \mu\text{A}/\text{cm}^2$. Similar temperature effects on device performance are also observed in TDNG with different friction layers such as Al-Kapton, PA6-Kapton, Cu-Kapton, Fe-Kapton and Al-PTFE. A wind-drive TDNG is demonstrated to power a temperature-humidity sensor, verifying the potential applications of such devices. Some important information is missing in the current version of the manuscript.

Response: We thank the reviewer very much for the positive and valuable comments. And we have carefully revised our manuscript according to these important comments to improve the quality of the manuscript.

1. Surface charge density is the key factor for high-performance triboelectric nanogenerators. Have the authors measured the surface charge density at different temperature differences? Is the surface charge density enhanced due to the change of temperature difference? Is the enhanced output current mainly attributed to the increase in surface charge density or other factors?

Response: Thank you very much for your comment. Surface charge density is the key factor for high performance TDNG. The surface charge density can be obtained by the ratio of the time integration of the current signal to the area of TDNG. As shown in **Fig. R10a**, as the temperature difference (ΔT) increases, the surface charge density of TDNG increases firstly and then decreases. When ΔT equals 145 K, the largest surface charge density can reach $58.8 \mu\text{C m}^{-2}$, which is 3 times that of ΔT equals 0 K ($19.6 \mu\text{C m}^{-2}$). In the last manuscript, this important information is reflected in the transferred short-circuit charge quantities per cycle (*CQC*, Figure 3a), which is more advantageous to show the practicability of triboelectric nanogenerators (*Energy Environ. Sci.* 2020; 13, 2069-2076). For clearer description, we have added the surface charge density to Supplementary Information.

In our work, the surface charge density is enhanced due to the change of temperature difference. The surface charge density of TDNG is the competition of the thermionic emission of the hotter part and the cooler part. Normally the increase of the hotter part's temperature (T_h) will increase the transferred charge density, which is beneficial to the charge accumulation of the cooler part. At the same time, the increase of the cooler part's temperature (T_c) will increase the charge loss due to the thermionic emission. These two factors compete with each other, resulting in a complicated trend of the surface charge density of the cooler part. For the hotter part, the higher the temperature, the stronger its thermionic emission capability, and the more transferred charge during the triboelectrification, which can be described as an empirical law showing the relationship of the transferred charge density (σ) and T_h (*Adv. Mater.* 2019; 31, e1808197):

$$\sigma = -C_1 T_h + C_2 \quad (\text{R8})$$

where σ is the transferred charge density when T_c keep fixed, C_1 and C_2 are the material-related correction factors. With T_h increases, the transferred charge density σ also increases, which enhances the surface charge density of the cooler part (σ_c). Simultaneously, T_c will rise with T_h increases due to the heat exchange in contact-separation processes and air heat transfer, and σ_c will follow an exponential decay during thermionic emission, which can be described as follows (*Adv. Mater.* 2018; 30, e1706790)

$$\sigma_c = e^{-SAt} \sigma_{c0} \quad (\text{R9})$$

where σ_{c0} is the initial value of σ_c , A is the surface area of TDNG, and t is the time of heat balance between the cooler part and hotter part, $S = \frac{\lambda_1 A_0}{k} T_c e^{\frac{qv}{kT_c}}$, where λ_1 is the material-specific correction factor, A_0 is Richardson constant of a free electron, k is Boltzmann constant. Based on the mentioned above, combining equation (R8) and equation (R9), the surface charge density of TDNG can be ultimately described as follows

$$\sigma_{TDNG} = (-C_1 T_h + C_2) e^{-SA t} \quad (R10)$$

In our experiments and simulations, due to the heat exchange in contact-separation processes and air heat transfer, T_c increases approximately linearly with T_h (as shown in the inset of Figure 1b), which can be approximately described as $T_c = aT_h + b$, where a and b are material and heat conduction related correction factors, which is related to the thermal conductivity of the materials of the cooler part and the heat transfer efficiency between the hotter part and the cooler part. In this way, T_c and T_h can be unified by temperature difference ($\Delta T = T_h - T_c$), as $(a\Delta T + b)/(1-a)$ and $(\Delta T + b)/(1-a)$. Substituting T_c and T_h into equation (R9), the relationship between the surface charge density of TDNG (σ_{TDNG}) and ΔT can be obtained, which can be described as follows

$$\sigma_{TDNG} = \left(-C_1 \frac{\Delta T + b}{1-a} + C_2\right) e^{-SA t} \quad (R11)$$

where $S = \frac{\lambda_1 A_0 (a\Delta T + b)}{k(1-a)} e^{\frac{qv(1-a)}{k(a\Delta T + b)}}$. Therefore, under the competition of the thermionic emission of the hotter part and the cooler part, the surface charge density of TDNG will increase first and then decreases as ΔT increases. In experiments, as shown in Figure 3, by controlling variables (such as driving frequency, contact separation speed, and driving force, *etc.*), with ΔT increases, the transferred short-circuit charge quantities per cycle, the accumulated charge in Kapton, and the surface potential of Kapton all increase first and then decrease, which directly indicates that the surface charge density in TDNG is enhanced due to the increase of ΔT .

The output performance of TDNG mainly depends on the change of surface charge density when keeping other factors constant (such as driving frequency, contact separation speed, and driving force, *etc.*). As shown in **Fig. R10b**, through the classical electrodynamics derivation and by ignoring edge effect, the voltage of TDNG (V) can be derived as follows

$$V = \frac{Q}{C} = -\frac{4\pi k_e \sigma_0}{\varepsilon \varepsilon_r} x(t) \quad (\text{R12})$$

where k_e is Coulomb's constant, σ_0 is the surface density of the Kapton friction layer, $x(t)$ is the distance between Kapton and Al friction layer, ε is the relative permittivity of Kapton, ε_r is the permittivity of vacuum. Combining equation (R11) and equation (R12), the relationship between the output performance of TDNG (V) and ΔT can be established as follows

$$V = \frac{Q}{C} = \frac{4\pi k_e \sigma_0}{\varepsilon \varepsilon_r} x(t) = -\frac{4\pi k_e x(t)}{\varepsilon \varepsilon_r} (-C_1 \frac{\Delta T + b}{1-a} + C_2) e^{-SAt} \quad (\text{R13})$$

From the equation, the output performance of triboelectric nanogenerator relates to the relative permittivity of Kapton (ε), the distance between Kapton and Al friction layer ($x(t)$, which relates the driving frequency, contact separation speed, driving force, *etc*), and the surface charge density of Kapton friction layer (σ_0). When keeping the relative permittivity of Kapton, driving frequency, contact separation speed, and driving force constant, the output performance of TDNG mainly depends on the change of surface charge density. The surface charge density σ_0 is mainly related to ΔT , resulting in the output performance of TDNG is mainly related to ΔT . With ΔT increases, the surface charge density σ_0 increases first then decreases, resulting in the output of TDNG increases first then decreases (Figure 2b, and Figure 5). This important information has been added to the manuscript and Supplementary Information.

Fig. R10 a The surface charge density at different temperature differences. **b** Schematic diagram of the relationship between voltage output (V) and surface density (σ_0).

2. In Equation (2): What is “A”? What are the material parameter values used in calculating the curve in Figure 1(b)?

Response: Thank you very much for your comment. In equation (2), “A” is the surface area of TDNG. The material parameters used in calculating the curve in Figure 1b are given in **Table R2**. When calculating the heat conduction (the right up inset in Figure 1b, the relationship between the temperature of the cooler part and hotter part), the heat capacity, thermal conductivity, and density of Al, Kapton, and air have been used. When calculating the change in surface charge density of the cooler part, the material-related correction factors in equation (1) of C_1 and C_2 equals 0.78 and 92, respectively (*Adv. Mater.* 2019; 31, e1808197). This important information has been added to the manuscript and Supplementary Information.

Table R2 The material parameters used in simulation and calculation.

Parameter	heat capacity (J kg ⁻¹ K ⁻¹)	thermal conductivity (W m ⁻¹ K ⁻¹)	density (kg m ⁻³)	relative permittivity
Al	900	238	2700	1
Kapton	1090	0.12	1430	3.7
Air	1.005	0.023	1.293	1.0005

3. What is the purpose of the Al film treatment including the immersing in 0.5 mol/L NaOH solution for 2 minutes and being washed by deionized water?

Response: Thank you very much for your comment. The Al film treatment of immersing in 0.5 mol L⁻¹ NaOH solution for 2 minutes is to prepare nanostructure on the surface of Al film for increasing the effective friction area. Being washed by deionized water is to remove impurities such as NaOH, Al(OH)₃, and NaAlO₂ remaining on the surface of the

Al film. As shown in **Fig. R11a**, the surface of the untreated Al film is relatively flat, and the friction area of the surface is small, which is not beneficial for triboelectrification. Through immersing Al film in 0.5 mol L⁻¹ NaOH solution, the following chemical reactions will occur

The generated hydrogen (H₂) on the surface of Al film will form tiny bubbles and adhere to the surface of the Al film, making the contact between the surface of the Al film and the NaOH solution uneven. The part of Al in contact with the NaOH solution will be corroded, and the part in contact with the H₂ bubbles will be retained. In this way, the nanostructures of Al can be prepared, and the nanostructures fabricated on Al film can enhance the friction area during triboelectrification to improve TDNG's output performance. Different immersing times can produce different nanostructures, as shown in **Fig. R11b-d**, when the Al film is immersed in 0.5 mol L⁻¹ NaOH solution for 1 minute, the relatively large-scale nanostructure can be produced; and when the immersing time increases from 1 min to 3 min, the size of the nanostructure will gradually become smaller and the contact area during triboelectrification will gradually become larger. However, the longer the reaction time, the worse the mechanical properties of Al film. After careful consideration, we choose 2 minutes immersing in 0.5 mol L⁻¹ NaOH solution to prepare nanostructure for enlarging the effective friction area and maintaining the mechanical properties of Al film.

Fig. R11 SEM images of the Al film with different immersing times. **a** SEM image of the untreated Al film. **b-d** SEM images of the Al film immersing in 0.5 mol L⁻¹ NaOH solution for 1, 2, and 3 minutes.

4. The Al film is placed on a hot plate and used as the hotter side of TDNG. What is the temperature (or the temperature range) of the hot plate? If the temperature of the Al film is raised 219 degree C above that of the cooler side (kept at 299 K), it is very likely that a thin oxide layer (Al₂O₃) is quickly formed on the surface of the Al film. This Al₂O₃ may serve as an insulating layer that could prevent charge transfer between the hotter side and the cooler side of the device. The proposed schematic in Figure 1 (a) ignores the existence and the possible role of this insulating layer, why?

Response: Thanks for the reviewer's comment. The temperature range of the hotter part in experiments is 289-522 K. The existence and the possible role of Al₂O₃ can be ignored in the schematic in Figure 1a, and the reasons are listed as below:

Firstly, Al is indeed reactive with atmospheric oxygen, and a thin passivation layer of Al₂O₃ (~ 4 nm thickness) forms on any exposed Al surface in a matter of hundreds of picoseconds even at room temperature (*Phys. Rev. Lett.* 1999; 82, 4866-4869). As the EDX mapping images of the untreated Al film in **Fig. R12a** show, there is already Al₂O₃ on the surface of untreated Al film, although the content is very low. Similarly, Al₂O₃ is also present on the surface of the Al friction layer that has just been prepared (**Fig. R12b**), but the content is very low. Secondly, during Al₂O₃ growth, large pressure variations occur that result in rapid diffusion of atoms in Al₂O₃. The large negative pressure contribution from electrostatic forces in Al₂O₃ is partially offset by the positive contribution of steric repulsion. This results in Al₂O₃ remaining largely under negative pressure, which causes aluminum to diffuse toward the surface and oxygen to diffuse toward the interior of the cluster. The diffusivity of aluminum is 30% to 60% higher than that of oxygen in the oxide (*Phys. Rev. Lett.* 1999; 82, 4866-4869). Therefore, the resistance at the thin Al₂O₃-Al interface is not as large as expected. Finally, charge emission from Al in air was only about 1% of that from Al₂O₃ (*Mater. Sci. Technol.* 1999; 15, 1454-1458), the existence of Al₂O₃ does not serve as an insulating layer that could prevent charge transfer between the hotter part and the cooler part of TDNG, it may promote triboelectrification between friction layers. Based on the above information, we think the existence and the possible role of Al₂O₃ can be ignored in the proposed

schematic in Figure 1a. Similar considerations also appear in previous works (*Nano Energy* 2014; 4, 150-156).

Fig. R12 EDX-mapping images of the untreated Al film and the prepared Al friction layer. **a** EDX-mapping images of the untreated Al film surface structure and its elemental mapping images of Al, and O. **b** EDX-mapping images of the Al film immersing in 0.5 mol L⁻¹ NaOH solution for 2 minutes surface structure and its elemental mapping images of Al-K, and O-K.

5. The Al film temperature is assumed to be the same as the setting temperature of the hot plate, which is reasonable. The cooler side of the device physical touches the Al film in one contact-separation cycle. How to monitoring the temperature of the cooler side of the device? How to ensure the temperature of the cooler side always remains at 299 K regardless of the rising temperature of the hotter side? Can the authors provide a photo (or a short video) of the experimental setup and show a real device is under test?

Response: We thank the reviewer for this comment. In our work, the temperature of Al film is measured by a thermocouple, and this temperature cannot be assumed to be the same as the set temperature of the thermostatic heater. When the thermodynamic

equilibrium between the cooler part and hotter part is reached, the temperature of the cooler part will not change, and the temperature of the cooler part is also measured by a thermocouple (as shown in **Supplementary Movie 1**). At the same time, the cooling system is used to keep the low temperature of the cooler part.

Because of the heat exchange in contact-separation processes and air heat transfer between the hotter part and cooler part, under non-ideal situations, we cannot ensure the temperature of the cooler side always remains at 299 K regardless of the rising temperature of the hotter side, only can reduce the temperature change of the cooler part as much as possible. This important information is mentioned in lines 69, page 3 of the manuscript as “However, in practical conditions, the temperature of the cooler friction layer will also rise in practical applications due to the heat exchange in contact-separation processes and air heat transfer, which will decrease the TENG’s output performance.”, and lines 90, page 4 of the manuscript as “Take the heat exchanges between hotter and cooler friction layers into account, the temperature of the cooler friction layer will continuously rise through air and contacting heat transfer, and the accumulated charges of cooler friction layer will gradually escape to air as well as the hotter friction layer by thermally stimulated discharging.”. Besides, the heat exchanges between hotter and cooler friction layers are considered in theoretical calculations and experiments. As shown in the inset in the right upper corner of Figure 1b, Figure 2b, and Figure 5a, the temperature of the cooler part (T_c) is specially marked. Such as the temperature changes in Figure 2b, as the temperature of the hotter part increases (from 289 K to 522 K), the temperature of the cooler part rises from 289 K to 303 K, with a changed temperature range of 14 K. Furthermore, the output performance is a competition of the thermionic emission of the hotter part and the cooler part. Normally the increase of T_h will increase the transferred charge density, which is beneficial to the charge accumulation of the cooler part. At the same time, the increase of T_c will increase the charge loss due to the thermionic emission. These two factors compete with each other, resulting in a complicated trend of the surface charge density of the cooler part. To clearly show the structure and working processes of TDNG, the schematic diagram of the working mechanism of TDNG is shown in **Fig. R13a**. The photos of the experimental setup of the real device under testing process are demonstrated in **Fig. R13b**, which corresponding to

the schematic diagrams in **Fig. R13a**, respectively, where the real device under test also can be found in **Supplementary Movie 1**. This important information has been added to the manuscript and Supplementary Information.

Fig. R13 The schematic diagram and the picture of TDNG. **a** The schematic working mechanism of TDNG. **b** The corresponding optical photographs of a working TDNG.

6. Can the author provide more details about their COMSOL simulation (for Figure 1(b) and Figure 1(c))? Such as material parameters, specific model that was used, device dimensions, etc.

Response: Thank you very much for your comment. The simulation in Figure 1b and Figure 1c is independent except using the same 2D geometric model (**Fig. R14a**). Because of the convergence difficulties and deviation of the COMSOL program when simulating slab materials with a length/thickness ratio larger than 100 (*Nano Res.* 2016; 9, 800-807), the thickness of the Kapton friction layer is set as 10 μm ; and the Al friction layer's thickness is set as 30 μm ; the width of Kapton friction layer and Al friction layer are set as 300 μm . The temperature change of the cooler part in the right up part of Figure 1b is obtained by steady simulation, which uses the module of heat transfer in solids and fluids in COMSOL. The thermal conductivity of Kapton and Al are set as 0.12 $\text{W m}^{-1}\text{K}^{-1}$ and 238 $\text{W m}^{-1}\text{K}^{-1}$, respectively. Additionally, the Kapton and Al are both wrapped in an air domain, whose thermal conductivity is 0.023 $\text{W m}^{-1}\text{K}^{-1}$. Besides, the heat capacity of

Kapton, Al, and air are set as $1090 \text{ J kg}^{-1}\text{K}^{-1}$, $900 \text{ J kg}^{-1}\text{K}^{-1}$, and $1.005 \text{ J kg}^{-1}\text{K}^{-1}$, respectively. The density of Kapton, Al, and air are set as 1430 kg m^{-3} , 2700 kg m^{-3} , 1.293 kg m^{-3} , respectively (as shown in **Table R3**). Through parametric sweeping, when the temperature of the hotter part changes from 293 K to 893 K, the temperature distribution (**Fig. R14b**) and the temperature change of the cooler part can be obtained (the inset in the right up corner in Figure 1b), and the temperature difference between the hotter part and cooler part can be obtained.

Table R3 The material parameters used in simulation and calculation.

Parameter	thickness (μm)	width (μm)	heat capacity ($\text{J kg}^{-1}\text{K}^{-1}$)	thermal conductivity ($\text{W m}^{-1}\text{K}^{-1}$)	density (kg m^{-3})	relative permittivity
Al	30	300	900	238	2700	1
Kapton	10	300	1090	0.12	1430	3.7
Air	-	-	1.005	0.023	1.293	1.0005

The potential distribution of TDNG in Figure 1c is obtained by a steady simulation, which uses the electrostatic module in COMSOL. In the simulation, the relative permittivity of Kapton and Al is set as 3.7 and 1, respectively. Using the temperature of the cooler part and the hotter part obtained in simulations, the surface charge density of the hotter part and cooler part can be calculated through equation (1) and equation (2), respectively. By changing the boundary conditions of the surface charge density of Al and Kapton, the potential distribution of TDNG under different temperature differences is simulated (Figure 1c).

Fig. R14 The model and temperature changes of TDNG. **a** The 2D geometric model of TDNG in COMSOL simulation. **b** The temperature distribution of TDNG with different temperature differences.

7. How was the $396 \mu\text{A}/\text{cm}^2$ obtained from Figure 4 (f)? Is it the peak value, or peak-to-peak value? Or is it the average value? What is the size of the Al-PTFE TENG device that was used for this test? What is the actually measured short-circuit current?

Response: Thank you very much for your comment. We carefully thought about the method of obtaining the largest output short-circuit current density in our manuscript, and found it is not right. And now we have obtained this value according to statistical analysis. In the last manuscript, the largest current density of $396 \mu\text{A cm}^{-2}$ in Figure 4f is the peak value obtained by drawing a red line as shown in **Fig. R15a**. The size of the Al-PTFE TDNG in Figure 4f is 1 cm^2 . According to 1st reviewer's comment, to make our result is comparable with other work, the data in Figure 4e-f are retested. The largest open-circuit voltage is 2110 V, and the average open-circuit voltage is 1805 V (**Fig. R15c**); the largest current can reach $670 \mu\text{A}$, and the average current is $480.3 \mu\text{A}$ (**Fig. R15d**) corresponding to an average current density of $480.3 \mu\text{A cm}^{-2}$ (the area of TDNG is 1 cm^2). For the

statistical analysis of 10 tested voltage and current signals, the average open-circuit voltage is 1697 ± 91.1 V, and the average short-circuit current is 443 ± 46.6 μA , corresponding to an average current density of 443 ± 46.6 $\mu\text{A cm}^{-2}$. Although the average applied pressure is just 63% of that in reference, the short-circuit current density is 26.6% larger than that of previous work (350 $\mu\text{A cm}^{-2}$, *ACS Nano* 2019; 13, 4640-4646). This important information has been added to the manuscript and the Supplementary Information.

Fig. R14 The output performance of TDNG. **a** The method of obtaining the largest current density in Figure 4f. **b** The short-circuit current corresponding to Figure 4f. **c** The open-circuit voltage of TDNG with similar situations to the reference. **d** The short-circuit current of TDNG with similar situations to the reference.

8. Figure 5 (a) does not show a clear picture of the application test. Can the author upload a short video of the wind-driven TDNG device in operation?

Response: Thank you very much for your comment. According to your nice comment, we have replaced the picture in Figure 5a with the pictures as shown in **Fig. R16**. The picture in **Fig. R16a** is a picture of a stationary wind-driven TDNG, and the picture in **Fig. R16b** shows the motion of wind-driven TDNG under the driving of wind to clearly demonstrate the working behavior of the wind-driven TDNG. Besides, a short video (**Supplementary Movie 3**) of the wind-driven TDNG device in operation taken by a high-speed camera is upload to demonstrate the working behavior of the wind-driven TDNG.

Fig. R16 The optical photographs of wind-driven TDNG. **a** The optical photograph of a stationary wind-driven TDNG. Scale bar is 2 cm. **b** The optical photograph of the working wind-driven TDNG under the wind.

Furthermore, to clearly demonstrate the working behavior of the wind-driven TDNG, the schematic diagram of the working mechanism of the wind-driven TDNG, and sequences pictures of the working behavior of an operational wind-driven TDNG taken by a high-speed camera is demonstrated in **Fig. R17a-b**, respectively. Subsequently, the potential distribution of wind-driven TDNG is simulated in COMSOL to better

understand the wind energy harvesting processes (**Fig. R17c**). The important information has been added to the manuscript and Supplementary Information.

Fig. R17 The schematic diagram and pictures of an operating wind-driven TDNG. **a** The schematic diagram of the working mechanism of the wind-driven TDNG. **b** Sequence optical photographs of an operating wind-driven TDNG taken by a high-speed camera. **c** The potential distribution of the operating wind-driven TDNG at different times.

9. The device performance is maximized when the temperature difference between the hotter and cooler friction layers is above 100 degree C. Can the authors suggest/propose a few practical scenarios in which the temperature difference can reliably be maintained at above 100 degree C?

Response: We thank the reviewer for the suggestion. In practical scenarios, the cooler part can be cooled by natural cold sources, such as the cold wind (~293 K, as shown in Figure 5), cold water (~283 K), frozen soil (~273 K), ice (~273 K), *etc.* The hotter part can be heated by natural heat sources and heat sources in life, such as pavements and roofs after solar exposure (~ 348 K), car engines (358-378 K), exhaust pipes (593-673 K), the surface of high-speed aircraft (505-810 K), *etc.* Under the cooling effect of the cold sources, the temperature of the cooling part can be well maintained near the temperature of the cold source. The temperature of the hotter part depends on the environment in which the TDNG is used. When the cold wind (~293 K) is used as the cold source in the cooler part and the exhaust pipes (393-673 K) is used as the heat source in the hotter part, the temperature difference can be maintained as 100-380 K. When the cold water (~283 K) is used as the cold source in the cooler part and the car engines (358-378 K) is used as the heat source in hotter part, the temperature difference can be maintained as 70-95 K.

Response to reviewer 3

Comments:

The authors attempted to optimize the temperature difference between friction layers for high output of triboelectric generators (TENGs). This ideal is novel and interesting based on simulation combining the electron-cloud-potential-well model for triboelectrification and the thermionic-emission model. However, the mechanistic analysis and the experiments of the TENG device were not carried out systematically and carefully. For example, the relationships among the working temperature, the temperature difference, and the outputs of TENGs have not been discussed adequately. Moreover, in practice, the influence of the heat conduction

between the friction layers on the output of TENGs has not been addressed carefully. Several key questions are left without clear answers, such as the rational of inducing temperature difference to increase the output, the influence of heat conduction on the output of the TENG, and how to maintain the temperature difference. Therefore, this manuscript is not suggested to be published in its present form. More specific comments are listed below.

Response: We appreciate the reviewer very much for the positive evaluation and the valuable comments. And according to these important comments, we have carefully improved our work and revised the manuscript. The detailed responses to these comments are listed below as the one-by-one responses to the reviewer's specific comments.

1. On page 4, the authors stated that “On the one hand, during friction, more electrons will hop from the hotter friction layer to cooler friction layer (middle part of Figure 1a) to enhance the charge density. On the other hand, the increased temperature of the cooler friction layer will discharge the electrons accumulated in the cooler friction layer (electrons are easier to escape out of the potential well, and get back to the hotter friction layer in contact processes or spill into air, as the right part of Figure 1a shows), and lead to reduce charge density and output performance of TENG.” This statement is suggested to provide some evidence.

Response: Thank you very much for your nice suggestion. We have added new theoretical evidence and experimental evidence to prove this statement in the revised manuscript at page 4.

For the statement of “On the one hand, during friction, more electrons will hop from the hotter friction layer to cooler friction layer (middle part of Figure 1a) to enhance the charge density.”. Theoretically, at absolute zero (0 K), the electrons will fill up all available energy states below the Fermi level (E_f) in the ideal condition. Otherwise, some electrons are elevated to the level above the E_f , which follows the Fermi-Dirac distribution. In TDNG, when the hotter part and the cooler part are in contact, electrons

locating at a high energy level in the hotter part will transit to the surface states of the hotter part through triboelectrification. The higher the temperature of the hotter part (T_h), the more electrons locating at high energy level, which results in more efficient triboelectrification and larger surface charge density. In experiments, as shown in Figure 3a, when the temperature difference increases from 0 K to 155 K, the transferred short-circuit charge quantities per cycle has been enhanced from 49 nC to 147 nC; in Figure 3b, when the temperature difference increases from 0 K to 155 K, the time integral of current increases; in Figure 3c, the surface potential of the cooler part increases from -21.7 V to -112.3 V when the temperature difference increases from 0 K to 155 K. These experimental phenomena directly illustrate that the cooler part's surface charge density increases with the temperature of the hotter part increases, and indirectly illustrate that more electrons hop from the hotter friction layer to cooler friction layer with the temperature of the hotter part increases.

Similar to the above analysis, when the hotter part and the cooler part are separated, the electrons entering into the surface of the cooler part can also escape because of the thermionic emission effect of triboelectric charges, that is “the increased temperature of the cooler friction layer will discharge the electrons accumulated in the cooler friction layer (electrons are easier to escape out of the potential well, and get back to the hotter friction layer in contact processes or spill into air, as the right part of Figure 1a shows), and reduce the charge density and output performance of TENG”. Besides, the working mechanism of the thermally stimulated discharge current (TSD) is based on the thermionic emission effect. Charge release is controlled by molecular movements, which depend on the temperature of materials. As the temperature of materials increases, molecular movements become violent, and the charge accumulated in materials can be discharged (*Polym. J.* 1971; 2, 173-191). In experiments, when the temperature of the cooler friction layer increases from 299 K to 303 K (Figure 3a), the transferred short-circuit charge quantities per cycle has been decreased from 147 nC to 99 nC; in Figure 3b, when the temperature of the cooler friction layer increases from 299 K to 303 K, the time integral of current decreases; in Figure 3c, the surface potential of the cooler part decreases from -112.3 V to -66.4 V when the temperature of the cooler friction layer increases from 299 K to 303 K. These experimental phenomena directly illustrate that the

increased temperature of the cooler friction layer will discharge the electrons accumulated in the cooler friction layer, and indirectly illustrate that electrons are easier to escape out of the potential well and get back to the hotter friction layer in contact processes or spill into air, as the right part of Figure 1a shows. In Figure 2b and Figure 5a, the output of TDNG directly illustrate that the increased temperature of the cooler friction layer (299 K to 303 K, and 293 K to 305 K, respectively) lead to reduce surface charge density and output performance of TENG. These experimental phenomena directly illustrate that electrons accumulated in the cooler friction layer will be discharged the temperature of the cooler part increases, and indirectly illustrate that electrons are easier to escape out of the potential well, and get back to the hotter friction layer in contact processes or spill into air with the temperature of the cooler part increases.

2. On Page 5, how to obtain the equation (1) and (2)? What are the basic assumptions for these equations? And what is the relationship between equation (1) and (2)? Moreover, these equations may have spelling mistakes, where ΔT is not included?

Response: Thank you very much for the comment. In order to describe the overall influences of high temperature and low temperature on the output of TENG, we need to account for the factors favorable and adverse to increase the surface charge density of TDNG together. The equation (1) is an empirical law obtained from the nanoscale's charge transfer process at different thermal conditions (*Adv. Mater.* 2019; 31, e1808197), which represent the factors favorable to increase the surface charge density of TDNG. Equation (2) is obtained from a thermionic-emission model (*Adv. Mater.* 2018; 30, e1706790), which represent the factors favorable to increase the surface charge density of TDNG. Only when both of them are taken into account can the correct behavior of TDNG be given. These equations have non-spelling mistakes, in the last manuscript, we have not established the equation of the relationship between output and temperature difference, so the ΔT is not included. In the revised manuscript, we have established an equation of the relationship between output and temperature difference:

$$V = -\frac{4\pi k_e x(t)}{\varepsilon \varepsilon_r} (-C_1 \frac{\Delta T + b}{1-a} + C_2) e^{-SA t} \quad (\text{R14})$$

where the basic assumptions and detailed derivation are shown below.

The temperature of friction layers will affect the thermionic emission ability of friction layers to influence the surface charge density of friction layers, and eventually affect the output performance of triboelectric nanogenerator. For the hotter part, the higher the temperature, the stronger its thermionic emission capability, and the more transferred charge during the triboelectrification, which can be described as an empirical law showing the relationship of the transferred charge density (σ) and T_h (*Adv. Mater.* 2019; 31, e1808197):

$$\sigma = -C_1 T_h + C_2 \quad (\text{R15})$$

where σ is the transferred charge density when the temperature of the cooler friction layer (T_c) keep fixed, C_1 and C_2 are the material-related correction factors. With T_h increases, the transferred charge density σ increases, which enhances the surface charge density of the cooler part (σ_c). At the same time, the temperature of the cooler part (T_c) will rise with T_h increases due to the heat exchange in contact-separation processes and air heat transfer, resulting in an exponential decay of surface charge density of the cooler part (σ_c) during thermionic emission, which can be described as follows (*Adv. Mater.* 2018; 30, e1706790)

$$\sigma_c = e^{-SA t} \sigma_{c0} \quad (\text{R16})$$

where σ_{c0} is the initial value of σ_c , A is the surface area of TDNG, and t is the time of heat balance of the cooler part and hotter part. $S = \frac{\lambda_1 A_0}{k} T_c e^{\frac{qv}{kT_c}}$, where λ_1 is the material-specific correction factor, A_0 is Richardson constant of a free electron, k is Boltzmann constant. Normally the increase of T_h will increase the transferred charge density, which is beneficial to the charge accumulation of the cooler part. Simultaneously, the increase of T_c will increase the charge loss due to the thermionic emission. These two factors compete with each other, resulting in a complicated trend of the surface charge density of the cooler part. Therefore, through combining equation (R15) and equation (R16), the surface charge density of TDNG can be described as follows

$$\sigma_{TDNG} = (-C_1 T_h + C_2) e^{-SA t} \quad (\text{R17})$$

In our work, T_c increases approximately linearly with T_h due to the heat exchange in contact-separation processes and air heat transfer, which can be described as $T_c = aT_h + b$, where a and b are material and heat conduction related correction factors, which are related to the thermal conductivity of the materials of the cooler part and the heat transfer efficiency between the hotter part and the cooler part. In this way, T_c and T_h can be unified by temperature difference ($\Delta T = T_h - T_c$), as $(a\Delta T + b)/(1-a)$ and $(\Delta T + b)/(1-a)$. And the surface charge density of TDNG (σ_{TDNG}) can be described as follows

$$\sigma_{TDNG} = \left(-C_1 \frac{\Delta T + b}{1-a} + C_2\right) e^{-SAt} \quad (R18)$$

where $S = \frac{\lambda_1 A_0 (a\Delta T + b)}{k(1-a)} e^{\frac{qv(1-a)}{k(a\Delta T + b)}}$. In this way, the relationship between the output performance of TDNG (V) and ΔT can be established as follows

$$V = \frac{Q}{C} = \frac{4\pi k_e \sigma_0}{\varepsilon \varepsilon_r} x(t) = -\frac{4\pi k_e x(t)}{\varepsilon \varepsilon_r} \left(-C_1 \frac{\Delta T + b}{1-a} + C_2\right) e^{-SAt} \quad (R19)$$

where k_e is Coulomb's constant, $x(t)$ is the distance between two friction layers, which is a function of time. ε is the relative permittivity of the cooler friction layer, and ε_r is the permittivity of vacuum. Therefore, under the competition of the thermionic emission of the hotter part and the cooler part, the surface charge density and the output of TDNG will increase first and then decrease as ΔT increases.

3. On page 8, the authors stated that ‘‘Furthermore, with the help of materials optimizing, the largest open-circuit voltage can reach to 1.5 kV (Figure 4e), and a largest short-circuit current density can reach to 396 $\mu\text{A}/\text{cm}^2$ (Figure 4f) for Al-PTFE TDNG, which is 13% larger than the record value (350 $\mu\text{A}/\text{cm}^2$).¹⁹’’. The repeatability of the peak outputs of voltage and current is normally weak. The authors utilized the largest peak current density to compare with the output of the reference. It seems that this comparison is unfair as well as the improvement is very slight. Because the variation of these measurements should be considered carefully. More importantly, such peak current density relies on the separation speed of the friction layers.

Response: Thank you very much for your comment. We carefully thought about the method of obtaining the largest output short-circuit current density in our last manuscript, and found it is not so reasonable. According to your comment, the output is retested with similar parameters in the compared reference and a statistical analysis is used to handle the experimental data to solve the problem of weak repeatability of the peak outputs of voltage and current. Besides, we have compared the current density of the compared reference to compare current density fairly, where the contact-separation velocity is also considered.

Considering that parameters we used in Figure 4f in the last manuscript (driving frequency ~ 4.4 Hz, applied pressure ~ 80 kPa) are different from those in the compared reference (driving frequency ~ 0.8 Hz, applied pressure 150 kPa (*ACS Nano* 2019; 13, 4640-4646)), the results in Figure 4e-f are retested with similar parameters in reference and replaced the previous ones in the manuscript (as shown in **Figure R18a-b**). Under the premise of 0.8 Hz driving frequency, the average applied pressure of 94.5 kPa (we cannot achieve an applied pressure of 150 kPa limited by the equipment), at the optimal temperature difference of 90 K, the largest open-circuit voltage is 2110 V, and the average open-circuit voltage is 1805 V (**Fig. R18a**); the largest short-circuit current of the PTFE-Al TDNG can reach 670 μA , and the average current is 480.3 μA corresponding to an average current density of 480.3 $\mu\text{A cm}^{-2}$ (the area of the PTFE-Al TDNG is 1 cm^2). After the statistical analysis of ten tested voltage and current results, the average short-circuit current is 443 ± 46.6 μA , corresponding to an average short-circuit current density of 443 ± 46.6 $\mu\text{A cm}^{-2}$. Although the average applied pressure is 63% that in reference (150 kPa), the short-circuit current density is 26.6% larger than that of previous work (350 $\mu\text{A cm}^{-2}$, *ACS Nano* 2019; 13, 4640-4646).

The peak short-circuit current density mainly relies on the contact-separation velocity of friction layers. The relationship between the contact-separation velocity and the output of TDNG has been systematically studied. As shown in **Fig. R18c-d**, through keeping driving frequency and applied pressure constant, as the contact-separation velocity increases from 30 mm s^{-1} to 350 mm s^{-1} , the open-circuit voltage increases from 35 V to 753 V. In **Fig. R18a-b**, the contact-separation velocity is ~ 1286 mm s^{-1} , and the

contact-separation velocity in the compared reference is larger than 1286 mm s^{-1} (which can be derived from the data of the amount of charge transferred. With similar output, the amount of transferred charge in the compared reference (20.2 nC cm^{-2}) is smaller than our experimental value ($150 \pm 17.8 \text{ nC cm}^{-2}$), indicating that the width of the current pulse is narrower than our work and the contact-separation velocity is larger than that used in our work).

Fig. R18 The retested output of TDNG and the relationship between output and contact-separation velocity. **a** The open-circuit voltage of TDNG with similar parameters to the reference. **b** The short-circuit current of TDNG with similar parameters to the reference. **c** The relationship between the open-circuit voltage and contact-separation velocity. **d** The relationship between the peak open-circuit voltage and contact-separation velocity.

4. As shown in Figures 2 and 3, the surface profile of the friction layer has not been quantitatively described. Only small surfaces were illustrated. And the temperature changes were up to 219 K. The mechanical properties of these friction layers had a huge change during the measurements. As well as the real contact area changed. However, the influence of the change of the real contact area on the output was not discussed in the manuscript. The authors are suggested to consider this phenomenon carefully.

Response: We appreciate the reviewer very much for the valuable comments. The width and height of Al nanostructure are about 100 nm, and the friction area of the chemical reactive etched 20- μm -thick aluminum (Al) foil increases by $\sim 87\%$ compared with that of untreated Al foil. The width and height of these irregularly Kapton nanopillars are about 150 nm and 100 nm, respectively. The friction area increases by $\sim 96\%$ compared with that of untreated Kapton film. A larger scale of Al friction layers ($\sim 20 \mu\text{m} \times 15 \mu\text{m}$) and Kapton friction layer ($\sim 40 \mu\text{m} \times 30 \mu\text{m}$) is demonstrated. The deformation of the friction layers are almost negligible, the volume strain of Kapton and Al are 0.14% and 1.4%, respectively, obtained from the simulation result. Compared to the 3.0 times surface charge enhancement brought by the joint effect of charge transfer and dissipation process under the optimal temperature difference, the influence of materials' deformation, and the change of the real contact area on the output can be ignored. The specific explanation is shown as below:

The friction layer in the hotter part is a chemical reactive etched 20- μm -thick aluminum (Al) foil. As shown in **Fig. R19a**, through immersing in 0.5 mol L⁻¹ NaOH solution for 2 minutes, irregularly convex nanostructures have been formed on the surface of Al foil (different areas of Al surface are illustrated in the right, middle, and left part of **Fig. R19a**, respectively). The width and height of these irregularly convex Al nanostructures are about 100 nm, and the friction area increases by $\sim 87\%$ compared with that of untreated Al foil (as shown in **Fig. R19e-g**). The friction layer in the cooler part is a reactive ion etched (RIE) 100- μm -thick Kapton film. As shown in **Fig. R19b**, after the etching process in RIE chamber RIE for 1 h with 3 sccm Ar and 7 sccm CF₄, 250 W input power, and irregularly nanopillars form on the surface of Kapton film (different areas of

Kapton surface are illustrated in the right, middle, and left part of **Fig. R19a**, respectively). The width and height of these irregularly Kapton nanopillars are about 150 nm and 100 nm, respectively (**Fig. R19d**). The friction area increases by ~96% compared with that of untreated Kapton film (as shown in **Fig. R19c, d, g**). These nanostructures fabricated on friction layers can enhance the contact area during triboelectrification and improve TDNG's electrical output performance.

Fig. R19 Characterization of the surface morphology of the friction layers. **a** Scanning electron microscope (SEM) images of different areas of the nanostructure on Al foil. **b** SEM images of different areas of the nanostructure on Kapton film. **c** Atomic force microscopy (AFM) topography of an untreated Kapton. **d** Atomic force microscopy

(AFM) topography of a reactive ion etched Kapton. **e** AFM topography of an untreated Al foil. **f** AFM topography of a chemical reactive etched Kapton. **g** Surface area of different friction layers. Scanning area, $1 \mu\text{m}^2$.

Conventionally, the mechanical properties of the friction layers will change with the temperatures, especially for polymer friction layers. In our experiments, polymer friction layers are taken as the cooler part in TDNG, the temperature change of the cooler part is very small (from 299 K to 313 K, the temperature change is just 14 K), and the mechanical changes are almost negligible. The metal friction layers are taken as the hotter part in TDNG and their temperature change is larger (from 299 K to 531 K, the temperature change is 232 K) compared with that of the cooler part, and deformation will be induced due to the thermal stress. To clearly illustrate the deformations of friction layers in our experiments, a simulation has been carried out with COMSOL, which uses a couple of the heat transfer in solids and fluids module and the solid mechanics module. In our simulation, the thermal expansion coefficient of Kapton and Al are set as $2.3 \times 10^{-5} \text{ K}^{-1}$ and $2 \times 10^{-5} \text{ K}^{-1}$, respectively. The thermal conductivity of Kapton and Al are set as $0.12 \text{ W m}^{-1}\text{K}^{-1}$ and $238 \text{ W m}^{-1}\text{K}^{-1}$, respectively. The Kapton and Al are both wrapped in an air domain, whose thermal conductivity is $0.023 \text{ W m}^{-1}\text{K}^{-1}$. Besides, the heat capacity of Kapton, Al, and air are set as $1090 \text{ J kg}^{-1}\text{K}^{-1}$, $900 \text{ J kg}^{-1}\text{K}^{-1}$, and $1.005 \text{ J kg}^{-1}\text{K}^{-1}$, respectively. The density of Kapton, Al, and air are set as 1430 kg m^{-3} , 2700 kg m^{-3} , 1.293 kg m^{-3} (as shown in **Table R4**), respectively. With the temperature of the hotter part (T_h) changes from 293 K to 523 K (the temperature change in experiments), the temperature of Kapton (T_c), volume strain, and strain displacement of Kapton and Al can be obtained. As shown in **Fig. R20a**, the temperature change of the cooler part is just 24 K with T_h increases from 293 K to 523 K. As shown in **Fig. R20b-c**, the deformation and displacement caused by the temperature change are 0.14%, $0.1 \mu\text{m}$ as for Kapton, and 1.4%, $3.9 \mu\text{m}$ as for Al. The deformation distribution of TDNG is demonstrated in **Fig. R20d**, compared to the whole friction layer, the deformation is very small. Therefore, in contrast to the 3.0 times surface charge enhancement brought by the joint effect of charge transfer and dissipation process under the optimal temperature difference, the influence of

materials' mechanical properties, deformation, and the change of the real contact area on the output can be ignored.

Table R4 The material parameters used in deformation simulation.

	thickness (μm)	width (μm)	heat capacity ($\text{J kg}^{-1}\text{K}^{-1}$)	thermal conductivity ($\text{W m}^{-1}\text{K}^{-1}$)	density (kg m^{-3})	thermal expansion coefficient (K^{-1})
Al	30	300	900	238	2700	2×10^{-5}
Kapton	10	300	1090	0.12	1430	2.3×10^{-5}
Air	-	-	1.005	0.023	1.293	-

Fig. R20 The strain of TDNG when temperature of the hotter part increases from 293 K to 523 K. **a** The relationship between the temperature of the cooler part and the temperature of the hotter part. **b** The strain of Al and Kapton when temperature of the hotter part increases from 293 K to 523 K. **c** The displacement of Al and Kapton when temperature of the hotter part increases from 293 K to 523 K. **d** The displacement distribution of TDNG under different temperature differences.

5. The size of the demonstration in Figure 5a should be given. The wind speed of 15 m/s is very large. More demonstrations based on smaller wind speed as common cases are suggested. Moreover, such an output is not surprising. Some benchmarkers from other references are suggested to be added for comparison.

Response: We appreciate the reviewer very much for the valuable comments. The size of the wind-driven TDNG is 12.6 cm \times 7 cm in Figure 5a. The optical photograph of the wind-driven TDNG has been replaced with a clearer one, and a scale bar of 2 cm has been added (**Fig. R21a**). Besides, to clearly demonstrate the working behavior of the wind-driven TDNG, the optical photograph was taken from a working wind-driven TDNG (**Fig. R21b**) is supplied and a short video (**Supplementary Movie 3**) of the working wind-driven TDNG is upload.

The wind speed of 15 m s⁻¹ is indeed very large. Therefore, we replaced the 100- μ m-thickness Kapton film with a 25- μ m-thickness Kapton film to make the wind-driven TDNG more suitable for low wind speed. In the new test, the wind speed is set to 5 m s⁻¹. When ΔT equals to 0 K, the largest open-circuit voltage and short-circuit current can reach 311 V and 47 μ A respectively, corresponding to an averagely transferred charge per unit time (Q) of 1.68 μ C (**Fig. R21c**). The optimum temperature difference is 68.7 K, under which the largest open-circuit voltage can reach 427 V, and the largest short-circuit current can reach 123.3 μ A, corresponding to an averagely transferred charge per unit time of 3.09 μ C. Under the optimal ΔT (68.7 K), the largest open-circuit voltage, largest short-circuit current, and averagely transferred charge in per unit time of the wind-driven TDNG are 1.37, 2.62, and 1.84 times the output values when ΔT equals to 0 K, respectively.

Some benchmarks from references are added for comparison. As **Table R5** shows, except for few works (*Nano Energy* 2019; 55, 260-268, *Nano Res.* 2018; 11, 1873-1882, the wind speed used for obtaining the largest output in these works are larger than 10 m s⁻¹), the largest short-circuit current of 123.3 μ A in our work is larger than previous works.

Fig. R21 The optical photographs and output of the wind-driven TDNG. **a** The optical photograph of a stationary wind-driven TDNG. Scale bar, 2 cm. **b** The optical photograph of an operating wind-driven TDNG. **c** The open-circuit voltage and the short-circuit current of the wind-driven TDNG under a wind speed of 5 m s^{-1} . Inset is the averaged transferred charge per unit time (Q).

Table R5 The electrical performance of previous flutter TENGs and our work.

voltage (V)	current (μA)	wind speed (m s^{-1})	References
60	2	10	Extreme Mech. Lett. 2017; 15, 122-129
98	16.3	27	Adv. Mater. 2016; 28, 1650-1656
50	45	22	ACS Nano 2016; 10, 1780-1787
396	75	18.4	Adv. Energy Mater. 2016; 6, 1501799
250	70	22	Nat. Commun. 2014; 5, 4929
36	11.8	17.9	Chem. Phys. Lett. 2016; 653, 96-100
135	12	24.6	Nano Energy 2017; 41, 210-216
140	1	15	Nano Energy 2017; 33, 418-426
63.3	15	20	Nano Energy 2017; 42, 269-281
50	15	10	Nano Energy 2016; 28, 288-295
21	-	25	Nano Energy 2015; 14, 201-208
1.25	0.14	3	
150	70	7	Nano Energy 2019; 55, 260-268
-	150	10	
-	12.5	15.8	Extreme Mech. Lett. 2018; 19, 46-53
1000	90	14	Nano Energy 2018; 53, 622-629
320	27	17	Adv. Mater. Technol. 2018; 3, 1700317
-	17.5	17.3	J. Mater. Chemi. A 2014; 2, 2079-2087
200	20	20	ACS Appl. Mater. Interfaces 2017; 9, 43716-43723
39	3	10	Nano Res. 2017; 11, 101-113
49	5	6.8	
375	248	14.5	Nano Res. 2018; 11, 1873-1882
-	326	20	
-	73	3	
110	80	10	ACS Nano 2013; 7, 9461-9468
10.7	-	2.7	Adv. Energy Mater. 2015; 5, 1501152
-	-	4.9	
-	24	18	J. Phys. D: Appl. Phys. 2016; 49, 215601
12	1.8	22.5	Sci. Rep. 2016; 6, 33977
427	123.3	5	This work

REVIEWER COMMENTS

Reviewer #1 (Remarks to the Author):

In this manuscript, authors reported a type of temperature difference triboelectric nanogenerator to enhance the electrical output performance in high temperature environment. As the hot part's temperature of nanogenerator is 0 K to 145 K higher than room temperature, the output voltage, current, surface charge density and output power are increased 2.7, 2.2, 3.0 and 2.9 times, respectively. The mechanism of the enhancement is discussed. It would be an interesting discovery for improvement of TENG in high temperature environment. The manuscript could be accepted after addressing the following questions.

1. In the experiment, authors use electric thermostat heater to keep higher temperature part. We have ever done such experiment. When the electric heater or electric cooler is used, the electrostatic induction charge has very strong influence on the surface of the tribolayer. Authors should do an experiment without using electric heater to prove the correction of the results. This is very important for this manuscript.
2. In Fig.1d, the temperature T_c of cooler part decreases with the thermal conductivity of the friction layer. As we known, the friction layer might get more heat from higher part in contact process if the thermal conductivity is higher for the friction layer. Authors should give detail explanation in this part.
3. In Table 1, I_o/I_r , here the I_o means the short circuit current? It would be changed into correct expression.
4. In supplementary Fig.5, why the charge on the Kapton film changes into positive charge in step iii?
5. In the demonstrations, 1882 LEDs are lighted by the TDNG, but they are not bright. Authors might cut half the LEDs to let them bright.

Reviewer #2 (Remarks to the Author):

The authors have addressed all my concerns/questions. I have no more comments.

Reviewer #3 (Remarks to the Author):

The quality of the revised manuscript has been improved. However, some descriptions seem not appropriate, discussions need further clarification and elaboration. They require attention before the manuscript is accepted for publication.

The detailed comments are listed below:

- (1) In reference 23, the original format of equation 2 in this manuscript is utilized to describe the short-circuit transfer charge Q_{sc} . Why did the authors change it to the surface charge density directly? Moreover, equation (1) is an empirical equation for another system with different materials. If the authors used this equation to explain the mechanism, some evidence is suggested to be added. The authors are suggested to verify if these frictions have the same trend on the change of surface charge density while the frictional materials are different.
- (2) The output performance of TDNG described by equation (4) is not appropriate for the case that when the TDNG works with a load. Equation (4) just describes the open-circuit voltage rather than a general output voltage with a load.

(3) The influence of temperature on the output performance is still not clear enough. For example, As shown in Figure 3c, when the temperature of the cooler friction layer increases from 299 K to 303 K, the transferred charge and the surface potential show a large change (147 nC to 99 nC, and -112.3 V to -66.4 V). What has happened in this small change of temperature? Moreover, the evidence listed by the authors is insufficient to support the argument of "Charge release is controlled by molecular movement, which depends on the temperature of materials."

(4) In the manuscript, the authors have stated that the surface area of the friction layers after treatment is increased. However, it cannot directly demonstrates that there is increased contact area since much surface area is not the real contact area.

(5) In the manuscript, the theoretical study in 2.1 has not been utilized to explain any experimental phenomena. Therefore, some discussions of figure 2 – 5 are suggested to be related to the theoretical study.

Response to the reviewers' comments

Response to reviewer 1

Comments to the Author

In this manuscript, authors reported a type of temperature difference triboelectric nanogenerator to enhance the electrical output performance in high temperature environment. As the hot part's temperature of nanogenerator is 0 K to 145 K higher than room temperature, the output voltage, current, surface charge density and output power are increased 2.7, 2.2, 3.0 and 2.9 times, respectively. The mechanism of the enhancement is discussed. It would be an interesting discovery for improvement of TENG in high temperature environment. The manuscript could be accepted after addressing the following questions.

Response: We appreciate very much the reviewer's very positive and valuable comments. And we have carefully revised our manuscript according to these important comments.

1. In the experiment, authors use electric thermostat heater to keep higher temperature part. We have ever done such experiment. When the electric heater or electric cooler is used, the electrostatic induction charge has very strong influence on the surface of the tribolayer. Authors should do an experiment without using electric heater to prove the correction of the results. This is very important for this manuscript.

Response: We thank you very much for your careful review and constructive suggestion. According to your suggestion, we did an experiment using a charcoal stove as a heating system and a water cooling system as a cooling system to avoid the influence of electrostatic induction. The results also illustrate that "the transferred charge density of TENG will first increase and then decrease when ΔT increases", which is consistent with the results/trend obtained by using electric heating system.

As shown in **Fig. R1 a-b**, we made a charcoal stove as a heating system consisting of a rectangular steel tube and asbestos. An insulating corundum plate is added between the friction layer and stove to avoid the possible electrostatic induction from the stove. When the burning charcoal is added into the stove, the highest temperature of the Al

friction layer can reach ~600 K. The cooler friction layer of the TDNG is cooled by a water cooling system without electrostatic induction. In this way, the output of TDNG under different temperature differences (ΔT) are obtained (**Fig. R1c**). As **Fig. R1d** demonstrated, when the temperature of the hotter friction layer T_h is 373.2 K (± 2 K), the temperature of the cooler friction layer T_c is 301.7 K, and ΔT is 71.5 K (± 2 K), the largest short-circuit current of TDNG can reach $26.49 \pm 0.36 \mu\text{A}$, which is 3.66 times of the output current at ΔT equals 3.5 K ($7.21 \pm 0.22 \mu\text{A}$). Besides, at this situation, as shown in **Fig. R1e**, the transferred charge per cycle is the maximum $112.18 \pm 6.88 \text{ nC}$, which is 1.76 times of the transferred charge per cycle $63.68 \pm 5.97 \mu\text{C}$ at temperature difference 3.5 K and cooler friction layer's temperature 294.2 K. The results illustrate that the transferred charge density of TDNG will first increase and then decrease when ΔT increases under the circumstance of avoiding electrostatic induction. This important information has been added in the manuscript and the Supplementary Information.

Fig. R1 The relationship between the output of TDNG and ΔT (ΔT is constructed using stove heated by charcoal not using the electric heating system). **a)** The schematic diagram of the TDNG using the stove as a heating system to increase the temperature of hotter friction layer and using the cooling water to keep the low

temperature of cooler friction layer. **b)** The optical photograph of the real TDNG using the stove as a heating system. **c)** The output current curve of TDNG with the increase of ΔT . **d)** The peak current of TDNG with the increase of ΔT . **e)** The transferred charge per cycle of TDNG with the increase of ΔT .

2. In Fig.1d, the temperature T_c of cooler part decreases with the thermal conductivity of the friction layer. As we known, the friction layer might get more heat from higher part in contact process if the thermal conductivity is higher for the friction layer. Authors should give detail explanation in this part.

Response: We thank the reviewer for this comment. Indeed, the higher the cooler friction layer's thermal conductivity is, the more heat the friction layer gets from hotter parts during the contact process. But the getting heat during contact process is small comparing with the heat dissipating ability of the cooler friction layer because it is directly attached to the cooling system. In TDNG, the cooler friction layer is closely attached to a water cooling system. The temperature of the cooler friction layer is strongly influenced by the cooling system. So, based on such a special design, the high thermal conductivity helps the cooler friction layer dissipate the heat getting from the hotter part during the contact process through the cooling system, so the temperature of cooler friction layer T_c decreases with the thermal conductivity of the friction layer. To further explain this point, we made a simulation. As shown in **Fig. R2**, when the temperature of the heating system and cooling system keep fixed at 593.15 K and 283.15 K, respectively, the temperature of the cooler friction layer is 396 K when the thermal conductivity of the cooler friction layer is $0.01 \text{ W}\cdot\text{m}^{-1}\text{K}^{-1}$, and the temperature of the cooler friction layer decreases to 277 K when the thermal conductivity of the cooler friction layer increases to $1 \text{ W}\cdot\text{m}^{-1}\text{K}^{-1}$. This important information has been added in the manuscript and Supplementary Information.

Fig. R2 The relationship between the cooler friction layer’s temperature T_c and its thermal conductivity when the temperatures of heating system and cooling system keep at 593.15 K and 283.15 K, respectively.

3. In Table 1, I_o/I_r , here the I_o means the short circuit current? It would be changed into correct expression.

Response: We thank you very much for your comment. We are very sorry for the puzzling description. We have changed them into the clearer expression as following. I_{opt} is the short-circuit current of the TDNG at the optimal ΔT , I_0 is the short-circuit current of the TDNG when ΔT equals 0 K. V_{opt} is the open-circuit voltage of the TDNG at the optimal ΔT , V_0 is the open-circuit voltage of the TDNG when ΔT equals 0 K. This important information has been added in line 247, page 9 of the revised manuscript as “ V_{opt} and I_{opt} are the open-circuit voltage and short-circuit current of the TDNG at the optimal ΔT , respectively. V_0 and I_0 are the open-circuit voltage and short-circuit current of the TDNG when ΔT equals 0 K, respectively.”. And the caption of **Table 1** has been revised to “The ratio V_{opt}/V_0 and the ratio I_{opt}/I_0 of TDNG composed of different materials with different optimum ΔT .” for clearer expression.

Table 1 The ratio V_{opt}/V_0 and the ratio I_{opt}/I_0 of TDNG composed of different materials with different optimum ΔT .

Materials	Al-Kapton	PA-6-	Cu-	Fe-Kapton	Al-PTFE

		Kapton	Kapton		
Optimum ΔT	145 K	144 K	145 K	143 K	90 K
V_{opt}/V_0	2.7	3.2	3.0	3.9	2.7
I_{opt}/I_0	2.2	2.0	2.6	10	1.7

4. In supplementary Fig.5, why the charge on the Kapton film changes into positive charge in step iii?

Response: We thank the reviewer for this comment. The charge on the Kapton film is negative. We are very sorry for this mistake in supplementary Fig.5. And we have replaced the wrong one in the Supplementary Information with the following revised figure (**Fig. R3**) as supplementary Fig.6.

Fig. R3 The working mechanism of TDNG. With a periodic change of working distance from I to IV, and IV to I, the amount of induced charges in Cr/Ag electrodes changes periodically, which produces an alternating current between electrodes.

5. In the demonstrations, 1882 LEDs are lighted by the TDNG, but they are not bright. Authors might cut half the LEDs to let them bright.

Response: We thank the reviewer for the nice comments. In accordance with your comment, we have cut half of the LEDs to let them bright. As shown in **Fig. R4**, 955 LEDs are lighted by a wind-driven TDNG. Compared with 1882 LEDs, the brightness of 955 LEDs significantly improved. This important information has been added in the manuscript and Supplementary Information.

Fig. R4 955 white LEDs are lightened by a wind-driven TDNG.

Response to reviewer 2

Comments:

The authors have addressed all my concerns/questions. I have no more comments.

Response: We appreciate the reviewer very much. And the manuscript has been carefully checked and revised again to further improve it.

Response to reviewer 3

Comments:

The quality of the revised manuscript has been improved. However, some descriptions seem not appropriate, discussions need further clarification and elaboration. They require attention before the manuscript is accepted for publication.

Response: We appreciate the reviewer very much for the positive evaluation and the valuable comments. According to these important comments, we have carefully improved our work and revised the manuscript. The detailed responses to these comments are listed below as the one-by-one responses to the reviewer's specific comments.

1. In reference 23, the original format of equation 2 in this manuscript is utilized to describe the short-circuit transfer charge Q_{sc} . Why did the authors change it to the surface charge density directly? Moreover, equation (1) is an empirical equation for another system with different materials. If the authors used this equation to explain the mechanism, some evidence is suggested to be added. The authors are suggested to verify if these frictions have the same trend on the change of surface charge density while the frictional materials are different.

Response: We thank you very much for your comment and suggestion. We will response to the comments in two parts separately. As for the comment “**In reference 23, the original format of equation 2 in this manuscript is utilized to describe the short-circuit transfer charge Q_{sc} . Why did the authors change it to the surface charge density directly?**”, we change the short-circuit transfer charge Q_{sc} to the surface charge density based on the following two reasons:

(1) There are three kinds of charge densities in TDNG, the surface charge density (σ_s), the transferred charge density in short-circuit condition (σ_{sc}), and the transferred charge density in the load-circuit condition (σ_l). Among these three kinds of charge densities, σ_s is more fundamental, based on which the transferred charge density can be obtained. σ_{sc}

and σ_t are the manifestation of electrostatic induced charges by σ_s during the movement of the friction layers. σ_t is related to σ_s according to the following equation:

$$\sigma_t = \sigma_s \left[1 - \left(1 + \frac{d}{RA\epsilon_0\epsilon_r} \int_0^t e^{\frac{1}{RA\epsilon_0\epsilon_r} \left(\frac{d}{\epsilon_r} t + \int_0^t x(t) dt \right)} dt \right) e^{-\frac{1}{RA\epsilon_0\epsilon_r} \left(\frac{d}{\epsilon_r} t + \int_0^t x(t) dt \right)} \right] \quad (1)$$

When R is equal to zero, the σ_{sc} is obtained naturally. According to Eq (1), when the two friction layers are far separated with the increase of t , σ_t is equal to σ_s . The derivation of equation (1) is given below:

In an ideal TDNG model (**Fig. R4a**, ignoring the influence of borders and heating system, and assuming that σ_s is distributed on the surface of the friction layer), the total charges in Al are $\sigma_s A - Q$ ($\sigma_s A$ is the triboelectric charges, and $-Q$ is the transferred charges between the two electrodes). The $V(t)$ - Q - $x(t)$ relationship for TDNG can be established as follows

$$V(t) = U_{Kapton} + U_{air} = -\frac{Q}{A\epsilon_0} \left(\frac{d}{\epsilon_r} + x(t) \right) + \frac{\sigma_s}{\epsilon_0} x(t) \quad (2)$$

where A is the surface area of TDNG, d is the thickness of the friction layer, ϵ_r is the permittivity of the Kapton, ϵ_0 is the permittivity of the vacuum, t is the time, and $x(t)$ is the distance between two friction layers. When the external load with resistance R is connected with TDNG, the output voltage of TDNG can be obtained by Ohm's law

$$V_{(t)} = I_{(t)} R = R \frac{dQ}{dt} \quad (3)$$

Combining equation (2) and (3), we can obtain

$$R \frac{dQ}{dt} + \frac{Q}{A\epsilon_0} \left(\frac{d}{\epsilon_r} + x(t) \right) = \frac{\sigma_s}{\epsilon_0} x(t) \quad (4)$$

When $t = 0$, the two friction layers are in contact, and there are no charges induced in electrodes ($Q = 0$). With this initial condition (when $t = 0$, $Q = 0$), equation (4) can be solved as follows

$$Q_{(t)} = \sigma_s S \left[1 - \left(1 + \frac{d}{RA\epsilon_0\epsilon_r} \int_0^t e^{\frac{1}{RA\epsilon_0\epsilon_r} \left(\frac{d}{\epsilon_r} t + \int_0^t x(t) dt \right)} dt \right) e^{-\frac{1}{RA\epsilon_0\epsilon_r} \left(\frac{d}{\epsilon_r} t + \int_0^t x(t) dt \right)} \right] \quad (5)$$

When R equals 1Ω , the transferred charge of the short-circuit current (Q) can be

calculated. As shown in **Fig. R4b**, with σ_s increases, Q increases, and Q changes linearly with σ_s (**Fig. R4c**). Divided by the surface area A , the relationship between σ_t and σ_s can be obtained.

$$\sigma_t = \sigma_s \left[1 - \left(1 + \frac{d}{RS\epsilon_0\epsilon_r} \int_0^t e^{\frac{1}{RS\epsilon_0} \left(\frac{d}{\epsilon_r} t + \int_0^t x(t) dt \right)} dt \right) e^{-\frac{1}{RS\epsilon_0} \left(\frac{d}{\epsilon_r} t + \int_0^t x(t) dt \right)} \right] \quad (6)$$

When the two frictions are far separated with the increase of t , σ_t is equal to σ_s , as shown in **Fig. R4d**.

(2) From the electron-cloud-potential-well model proposed in reference 23 (*Adv. Mater.* 2018, 30, 1706790), the charge transfer occurs in the contact process of the friction layers and the temperature of friction layers affects the short-circuit transfer charge Q_{sc} . When two friction layers contact each other, the initial single potential well becomes asymmetric double-well potential and then the electron could hop from friction layer A to friction layer B. As the temperature increases, the energy fluctuations of charge become larger, and the charge is easier to hop out from the trap and go back to friction layer A, leading to a smaller Q_{sc} and a smaller quantity of accumulated charge. Besides the charge transfer in the contact process, our work found that the accumulated charge in friction layers will emit into air in the separation process (the right part in Fig. 1a), resulting in smaller σ_s . In summary, for TDNG, the temperature affects σ_s of the friction layer in the whole friction process, and the short-circuit transfer charge Q_{sc} should be changed to σ_s to describe the charge change in the whole friction process (include the contact process and the separation process) more comprehensively.

Fig. R4 The relationship between the transferred charge density and the surface charge density. a) The theoretical model of TDNG. **b-c)** The relationship between the transferred charge Q and surface charge density. **d)** The relationship between the transferred charge density and surface charge density.

As for the comment “Moreover, equation (1) is an empirical equation for another system with different materials. If the authors used this equation to explain the mechanism, some evidence is suggested to be added. The authors are suggested to verify if these frictions have the same trend on the change of surface charge density while the frictional materials are different.”, these frictions do have the same trend on the change of surface charge density while the frictional materials are changed to Al-Kapton in our work. It can be verified through the following discussion.

The relationship between the transferred charge density and the temperature of the hotter friction layer (T_h) can be gained by testing the change of the surface potential of the cooler friction layer with increasing T_h in experiments (*Sensor Actuat. A-Phys.* 1992, 32, 357-360). As same as reference 24 (*Adv. Mater.* 2019, 31, 1808197), in the range of 313 K to 433 K, a linear change between the surface potential of the cooler friction layer and T_h has been observed (**Fig. R5a**). The surface potential is tested with an electrostatic voltmeter (Monroe ME-279/220V). Ignoring the influence of the border, the probe of the electrostatic voltmeter and the friction layer make up a capacitor C , which can be described as follows

$$C = \frac{Q}{U} \quad (7)$$

where Q is the accumulated charge on the surface of the friction layer, U is the potential drop between the probe and the surface of the friction layer, which equals to the surface potential (V_s) of the friction layer when the potential of the probe is zero. In this case, the relationship between the surface charge density and surface potential can be established as follows

$$\frac{\varepsilon S}{4\pi k d_0} = \frac{\sigma_s S}{V_s} \quad (8)$$

$$\sigma_s = \frac{\varepsilon V_s}{4\pi k d_0} \quad (9)$$

When the friction layers are far separated, the surface charge density σ_s equals the transferred charge density σ_t . Therefore, the relationship between σ_t and surface potential can be formulated as follows

$$\sigma_t = \frac{\varepsilon V_s}{4\pi k d_0} \quad (10)$$

As shown in **Fig. R5b**, σ_t is almost linear with T_h . This phenomenon illustrates that the metal-polymer friction system has the same trend as equation (1) in our manuscript (cited from reference 24, *Adv. Mater.* 2019, 31, 1808197).

Fig. R5 The relationship between the surface potential of the cooler friction layer of TDNG and T_h . **a)** The relationship between the surface potential of the cooler friction layer of TDNG and T_h . **b)** The relationship between the transferred charge density of TDNG and T_h .

2. The output performance of TDNG described by equation (4) is not appropriate for the case that when the TDNG works with a load. Equation (4) just describes the open-circuit voltage rather than a general output voltage with a load.

Response: We thank you very much for the comment. According to your comments, we have rebuilt the following equation of the output voltage of TDNG to make it appropriate for the case when the TDNG works with external loads.

$$V(t) = \frac{(-C_1 \frac{\Delta T + b}{1-a} + C_2) d e^{-SA t_0}}{\epsilon_0 \epsilon_r} \left[(d + x(t) \epsilon_r) \left(\frac{1}{d} + \frac{\int_0^t e^{\frac{1}{RA\epsilon_0} (\frac{d}{\epsilon_r} t + \int_0^t x(t) dt)} dt}{RA\epsilon_0 \epsilon_r} \right) e^{-\frac{1}{RA\epsilon_0} (\frac{d}{\epsilon_r} t + \int_0^t x(t) dt)} - 1 \right] \quad (11)$$

The equation in manuscript has been revised correspondingly. The basic assumptions and detailed derivation of above equation are given below.

As shown in **Fig. R7a**, an ideal TDNG model is built by ignoring the influence of borders and heating system and assuming that all transferred charges are concentrated on the surface of the friction layer. The total charges in Al are $\sigma_s A - Q$ ($\sigma_s A$ is the triboelectric charges, and $-Q$ is the transferred charges between the two electrodes). The $V(t)$ - Q - $x(t)$ relationship for TDNG can be established as follows

$$V_{(t)} = U_{Kapton} + U_{air} = -\frac{Q}{A\varepsilon_0} \left(\frac{d}{\varepsilon_r} + x_{(t)} \right) + \frac{\sigma_{TDNG}}{\varepsilon_0} x_{(t)} \quad (12)$$

where A is the surface area of TDNG, d is the thickness of the friction layer, ε_r is the permittivity of the Kapton, ε_0 is the permittivity of the vacuum, t is the time, and $x_{(t)}$ is the distance between two friction layers. When the external load with resistance R is connected with TDNG, the output voltage of TDNG can be obtained by Ohm's law

$$V_{(t)} = I_{(t)}R = R \frac{dQ}{dt} \quad (13)$$

Combining equation (12) and (13), we can obtain

$$R \frac{dQ}{dt} + \frac{Q}{A\varepsilon_0} \left(\frac{d}{\varepsilon_r} + x_{(t)} \right) = \frac{\sigma_{TDNG}}{\varepsilon_0} x_{(t)} \quad (14)$$

When $t = 0$, the two friction layers are in contact, and there are no charges induced in electrodes ($Q = 0$). With this initial condition (when $t = 0$, $Q = 0$), equation (4) can be solved as follows

$$Q_{(t)} = \sigma_{TDNG} A \left[1 - \left(1 + \frac{d}{RA\varepsilon_0\varepsilon_r} \int_0^t e^{\frac{1}{RA\varepsilon_0} \left(\frac{d}{\varepsilon_r} t + \int_0^t x_{(t)} dt \right)} dt \right) e^{-\frac{1}{RA\varepsilon_0} \left(\frac{d}{\varepsilon_r} t + \int_0^t x_{(t)} dt \right)} \right] \quad (15)$$

The current output can be obtained by taking the first derivative of $Q(t)$ with respect to t .

$$I_{(t)} = \frac{\sigma_{TDNG} d}{R\varepsilon_0\varepsilon_r} \left[(d + x_{(t)}\varepsilon_r) \left(\frac{1}{d} + \frac{\int_0^t e^{\frac{1}{RA\varepsilon_0} \left(\frac{d}{\varepsilon_r} t + \int_0^t x_{(t)} dt \right)} dt}{RA\varepsilon_0\varepsilon_r} \right) e^{-\frac{1}{RA\varepsilon_0} \left(\frac{d}{\varepsilon_r} t + \int_0^t x_{(t)} dt \right)} - 1 \right] \quad (16)$$

Multiplying the current output $I(t)$ with the external resistance R , the voltage output across the external load can be obtained

$$V_{(t)} = \frac{\sigma_{TDNG} d}{\varepsilon_0\varepsilon_r} \left[(d + x_{(t)}\varepsilon_r) \left(\frac{1}{d} + \frac{\int_0^t e^{\frac{1}{RA\varepsilon_0} \left(\frac{d}{\varepsilon_r} t + \int_0^t x_{(t)} dt \right)} dt}{RA\varepsilon_0\varepsilon_r} \right) e^{-\frac{1}{RA\varepsilon_0} \left(\frac{d}{\varepsilon_r} t + \int_0^t x_{(t)} dt \right)} - 1 \right] \quad (17)$$

With the friction layers are separated linearly with time (same as that in our experiments):

$$x_{(t)} = vt \quad (18)$$

$$t \leq \frac{x_{max}}{v} \quad (19)$$

where v is the driving velocity, x_{max} is the maximum distance between two friction layers. The specific voltage and current of TDNG with different external loads can be obtained. As shown in **Fig. R7b-c**, with R increases from 1 k Ω to 1 G Ω , the peak voltage increases, and the wave crest's width gradually increases; the peak current decreases and the wave crest's width gradually increases. This is because Q can still get its saturation value when the top electrode stops moving ($t < 20$ ms) for a relatively small R . However, when R is more than 100 M Ω , at $t = 20$ ms, the charge cannot get saturated due to the limit charge transfer rate by the resistor, resulting in the unstopped charge transfer from the electrode to Al when $t > 20$ ms. Besides, according to the relationship between the transferred charge density and temperature difference

$$\sigma_{TDNG} = (-C_1 \frac{\Delta T + b}{1-a} + C_2) e^{-SA t_0} \quad (20)$$

the equation of voltage and current with external load can be described as follows

$$V(t) = \frac{(-C_1 \frac{\Delta T + b}{1-a} + C_2) d e^{-SA t_0}}{\varepsilon_0 \varepsilon_r} \left[(d + x(t) \varepsilon_r) \left(\frac{1}{d} + \frac{\int_0^t e^{\frac{1}{RA \varepsilon_0} (\frac{d}{\varepsilon_r} t + \int_0^t x(t) dt)} dt}{RA \varepsilon_0 \varepsilon_r} \right) e^{-\frac{1}{RA \varepsilon_0} (\frac{d}{\varepsilon_r} t + \int_0^t x(t) dt)} - 1 \right] \quad (21)$$

$$I(t) = \frac{(-C_1 \frac{\Delta T + b}{1-a} + C_2) d e^{-SA t_0}}{R \varepsilon_0 \varepsilon_r} \left[(d + x(t) \varepsilon_r) \left(\frac{1}{d} + \frac{\int_0^t e^{\frac{1}{RA \varepsilon_0} (\frac{d}{\varepsilon_r} t + \int_0^t x(t) dt)} dt}{RA \varepsilon_0 \varepsilon_r} \right) e^{-\frac{1}{RA \varepsilon_0} (\frac{d}{\varepsilon_r} t + \int_0^t x(t) dt)} - 1 \right] \quad (22)$$

As shown in **Fig. R7d-e**, with ΔT increases, the voltage and current output of TDNG with different external loads increases firstly and then decreases, which is consistent with the experimental results (Fig. 4a in manuscript). Besides, the output power of TDNG also increases firstly and then decreases as ΔT increases (**Fig. R7f**).

Fig. R7 The output voltage, current, and power of TDNG with different external loads and different ΔT . **a)** The theoretical model of TDNG. **b)** The theoretical output current of TDNG with different external loads in theory. **c)** The output voltage of TDNG with different external loads in theory. **d)** The theoretical output voltage of TDNG with different external loads and different ΔT . **e)** The theoretical output current of TDNG with different external loads and different ΔT . **f)** The theoretically largest output power of TDNG with different ΔT .

3. The influence of temperature on the output performance is still not clear enough. For example, as shown in Figure 3c, when the temperature of the cooler friction layer increases from 299 K to 303 K, the transferred charge and the surface potential show a large change (147 nC to 99 nC, and -112.3 V to -66.4 V). What has happened in this small change of temperature? Moreover, the evidence listed by the authors is insufficient to support the argument of “Charge release is controlled by molecular movement, which depends on the temperature of materials.”.

Response: We thank you very much for your comment. The influence of temperature on the output performance is discussed in more detail by describing the influence of T_h and T_c (**Fig. R8a-b**) respectively in the following. The transferred charge and the surface potential show a large change (147 nC to 99 nC, and -112.3 V to -66.4 V) when the temperature of the cooler friction layer increases from 299 K to 303 K can be explained by the two points below: **1)** the exponential decay of surface charge with T_c , as shown in equation (23):

$$Q = Q_0 A e^{-\frac{\lambda_1 \Lambda_0 T_c}{k} e^{\frac{q\psi}{kT_c A t_0}}} \quad (23)$$

When T_c reaches a certain value (299 K in our experiment), the surface charge as well as the surface potential which is linear with surface charge density on the cooler friction layer will quickly dissipate (exponential decay), resulting in a large change with a small increase of T_c . **2)** The whole friction process can be divided into the contact process and the separation process. In the contact process, the temperature of the surface of the cooler friction layer (within the thickness of dozens of atomic layers under the surface) may rise by dozens of Kelvin during the contact process (within tens of milliseconds). The thermionic emission of electrons becomes violent during the contact process, which leads to a big charge dissipation of the friction layer and a substantial decrease of output. Besides, the statement “Charge release is controlled by molecular movement, which depends on the temperature of materials.” that we used before is not rigorous enough. To be more rigorous, this sentence has been changed to “The charge release is controlled by

molecular movement and electron motions, which depends on the temperature of materials”. The detailed explanations of mentioned above are given below.

(1). The detailed response to the comment “**The influence of temperature on the output performance is still not clear enough.**”

The influence of temperature on the output performance is illustrated by numerical calculation (**Fig. R8a-b**). Both the temperature of the hotter friction layer (T_h) and the temperature of the cooler friction layer (T_c) influence the output of TDNG. The influence of T_h can be described by equation (24):

$$\sigma_t = -C_1 T_h + C_2 \quad (24)$$

According to equation (24), as T_h increases, the more electrons emitted from the hotter friction layer, which is beneficial to the transferred charge density (**Fig. R8a**). However, the increase of T_c decreases the accumulated charge on the cooler friction layer, which can be described by equation (25):

$$Q = Q_0 A e^{-\frac{\lambda_1 A_0}{k} T_c} e^{\frac{qv}{k T_c} A t_0} \quad (25)$$

According to equation (25), as T_c increases, the more electrons escape from the cooler friction layer, which decreases the accumulated charge as well as the surface charge density (where the surface charge density can be approximated as the ratio between the accumulated charge and surface area, that is, $\sigma_s = Q/S$). As shown in **Fig. R8b**, when T_c increases from 350 K to 370 K, the accumulated charge in the cooler friction layer decreases from -134 μC to 128 μC , and this trend will become more obvious as T_c becomes larger. Certainly, if we just consider the change of T_c or T_h , it can't fully describe the change of output with temperature. By combining equation (24) and equation (25), the overall influences of T_h and T_c on the output of TDNG can be described. The relationship between the surface charge density of TDNG (σ_{TDNG}) and ΔT can be described as follows

$$\sigma_{TDNG} = (-C_1 \frac{\Delta T + b}{1-a} + C_2) e^{-S A t_0} \quad (26)$$

where $S = \frac{\lambda_1 A_0 (a\Delta T + b)}{k(1-a)} e^{\frac{qv(1-a)}{k(a\Delta T + b)}}$, a and b are material and heat conduction related correction factors, which are related to the thermal conductivity of friction materials. As shown in **Fig. R8c**, the transferred charge density and surface potential of TENG will increase and then decrease. Besides, the relationship between the output and temperature has been summarized into equations (21) and (22) above to demonstrate the influence of temperature on the output performance. This important information has been added in the manuscript and Supplementary Information.

(2). The detailed response to the comment **“For example, as shown in Figure 3c, when the temperature of the cooler friction layer increases from 299 K to 303 K, the transferred charge and the surface potential show a large change (147 nC to 99 nC, and -112.3 V to -66.4 V). What has happened in this small change of temperature?”**

Due to the exponential decay of accumulated charge with T_c (equation (25)), when T_c reaches a certain value (299 K in our experiment), the surface charge density as well as the surface potential V_s that is linear with surface charge density (equation (9)) will exponentially decay, which leads to a large change of V_s with a small increase of T_c (**Fig. R8b**). Therefore, the transferred charge and the surface potential may show a large change in our experiments (147 nC to 99 nC, and -112.3 V to -66.4 V) when the temperature of the cooler friction layer increases from 299 K to 303 K. Besides, considering the more detailed working process of TDNG, the whole friction process includes the contact process and the separation process. Both the temperature of the cooler friction layer and the temperature of the hotter friction layer change in the dynamic contact and separation processes. The temperature of the surface of the cooler friction layer (within the thickness of dozens of atomic layers on the surface) may rise by dozens of Kelvin during the contact process (within tens of milliseconds). The thermionic emission of electrons increases quickly during the contact process, which leads to a significant increase in the charge dissipation of the friction layer. Simultaneously, the temperature of the surface of the hotter friction layer (within the thickness of dozens of atomic layers on the surface) may decrease by dozens of Kelvin during the contact process, which reduces the charge transferring. Moreover, as the temperature difference increases, the gradient of temperature between the hotter friction layer and the cooler

friction layer becomes larger, and this phenomenon will become more obvious. However, T_h and T_c measured in experiment is the average temperature at the whole friction process (that is, the temperature after thermal equilibrium). And equation (26) gives the effect of average temperature on TDNG not the real dynamic temperature of T_c and T_h during the whole friction process (including the contact process and the separation process). Therefore, because of the quick increase of T_c in the contact process and the exponential decay of accumulated charge with T_c , the transferred charge and the surface potential show a large change in our experiments (147 nC to 99 nC, and -112.3 V to -66.4 V) when the temperature of the cooler friction layer increases from 299 K to 303 K. We thank the reviewer again for the suggestion, and we will deeply study the influence of temperature in the contact process and the separation process on TENG in the future works.

Fig. R8 The relationship between transferred charge, accumulated charge, surface charge density and T_h , T_c , and ΔT . a) The relationship between transferred charge and

T_h in theory. **b)** The relationship between accumulated charge in cooler friction layer and T_c in theory. **c)** Numerical simulations of the relationship between ΔT and the surface charge density. The right upper is the relationship between the temperature of the hotter friction layer and the cooler friction layer.

(3). The detailed response to the reviewer's comment **“Moreover, the evidence listed by the authors is insufficient to support the argument of “Charge release is controlled by molecular movement, which depends on the temperature of materials.””**

The statement “Charge release is controlled by molecular movement, which depends on the temperature of materials.” that we used in the last response is not rigorous enough. To be more rigorous, it has been revised as “The charge release is controlled by molecular movement and electron motions, which depends on the temperature of materials”. Many studies show that the charge release is controlled by the collective effect of molecular movements and electron motions (*Polymer* 2003, 44, 2781-2791; *Polymer* 1994, 35, 1915-1922), where the molecular movements follow molecular dynamics and the electron motions follow electron thermal emission. For molecular movements, according to Brown theory, there are three types of molecular movements in an amorphous polymer (*Philos. Mag.* 1973, 18, 483; *J. Mater. Sci.* 1983, 18, 2241): 1) shearons which consist of the motion of molecular segments whose covalent bonds lie in the plane of shear; 2) rotons which are like shearons except that covalent bonds make an angle with the plane of shear; 3) tubons that require a force parallel to the covalent bond that allows the molecular segments to move along the shear plane. These molecular movements all depend on the temperature of materials, as the temperature increases, the thermal agitation can strain and release molecular chemical bonds via rotation, bond length, and bond angle movements. There are two kinds of charges in a charged polymer: dipole charge, space charge, where the dipoles charge connects with molecular movements. As the temperature increases, the molecular movements become violent, and the dipoles reorientate and the dipole charges release. Besides, based on previous works, for relaxations due to molecular movements of polar side groups, the Arrhenius equation is appropriate (*Polymer Journal*, 1971, 2, 173-191):

$$\alpha = \alpha_0 e^{-\frac{E}{kT}} \quad (27)$$

where α is the relaxation frequency, α_0 is an approximately constant, E is the activation energy, and k is the Boltzmann constant. As temperature T increases, the relaxation frequency increases, and the dipoles reorientate more easily (*Polymer Journal*, 1971, 2, 173-191). Therefore, the charge release is controlled by molecular movements, which depends on the temperature of materials. For electron motions, in a charged polymer, the space charge can be approximated as frozen in the trap of polymer at low temperatures, but when the temperature increases, they are remobilized. According to previous works, for a trap depth U , the escape rate ν of the space charge obeys Boltzmann statistic (Electrets, J. van Turnhout, 1987, chapter 3; Electrets, Charge Storage, and Transport in Dielectrics, Martin M. Perlman, 1973, chapter 6):

$$\nu(T) = \nu_0 e^{-\frac{U}{kT}} \quad (28)$$

The higher the temperature T , the greater the escape rate ν .

Based on mentioned above, the amount of released charge dQ in the time dt is proportional to the amount of charge Q accumulated in polymer:

$$-dQ = \nu_{(T)} Q dt \quad (29)$$

As t equals 0, the accumulated charge in the polymer is Q_0 . The remaining charge in polymer can be solved as

$$Q = Q_0 e^{-\nu_{(T)} t} \quad (30)$$

Combining equation (28), equation (30) and assuming T is proportional to time t ($T = t/a$, where a is a proportional constant), we can obtain

$$Q = Q_0 e^{-\nu_0 a T e^{-\frac{U}{kT}}} \quad (31)$$

And the released charge by thermally discharging can be obtained

$$Q_{dis} = Q_0 (1 - e^{-\nu_0 a T e^{-\frac{U}{kT}}}) \quad (32)$$

The thermally discharged current can be obtained

$$I_{ele} = \frac{dQ_{dis}}{dt} = v_0 a Q_0^2 e^{-v_0 a T} e^{-\frac{U}{kT}} e^{-\frac{U}{kT}} \left(1 + \frac{U}{kT}\right) \quad (33)$$

According to equation (33), the charge release controlled by electron thermal emission is demonstrated in **Fig. R9a**. Based on the similar derivation of the charge release controlled by electron motions, the charge release controlled by molecular movements can be established as follows

$$I_{mol} = \alpha_0 a Q_0^2 e^{-\alpha_0 a T} e^{-\frac{E}{kT}} e^{-\frac{E}{kT}} \left(1 + \frac{E}{kT}\right) \quad (34)$$

Besides, assuming the charge release is controlled by the molecular movements and electron motions simultaneously, and the total thermally discharge current can be obtained

$$I_{tot} = I_{ele} + I_{mol} = v_0 a Q_0^2 e^{-v_0 a T} e^{-\frac{U}{kT}} e^{-\frac{U}{kT}} \left(1 + \frac{U}{kT}\right) + \alpha_0 a Q_0^2 e^{-\alpha_0 a T} e^{-\frac{E}{kT}} e^{-\frac{E}{kT}} \left(1 + \frac{E}{kT}\right) \quad (35)$$

As shown in **Fig. R9b**, the total thermally stimulated discharge current has two peaks. The molecular movements result in the first current peak, and the electron motions result in the second current peak. In TSD current test, more than one current peaks appear (**Fig. R9c**, *J. Optoelectron. Adv. Mater.* 2006, 8, 962-966), and this phenomenon cannot be simply explained by thermal electron emission, where only one current peak can be observed (**Fig. R9a**). So, it can be inferred that some portion of charge release is controlled by molecular movements. When the temperature of the sample (a charged Teflon FEP) increases from 100 K to 200 K, the sharp molecular movements in the crystalline part of the polymer lead to two sharp TSD current peaks (γ_a and γ_c in **Fig. R9c**). Through molecular movements, the dipoles reorientate and the dipole charges release quickly. When the temperature of the sample increases from 200 K to 320 K, molecular movements in the amorphous part of the polymer lead to a decrease of trap depth, and electrons escape from the trap more easily, resulting in two broad TSD current peaks (β_a and β_c in **Fig. R9c**). The similar results have been reported in other studies (*Nano Energy* 2017, 37, 268-274; *AIP Advances* 2019, 9, 125334; *Smart Mater. Struct.* 2017, 26, 085001; *Sensors and Actuators A*, 1992, 32, 357-360). Besides, the results we obtained from the experiments (Fig. 3b) also reflect that the temperature change leads to a change of accumulated charge (the time integral of TSD current) in TDNG. Based on

the above discussion, the temperature determines the molecular movements and electron motions, so as to further control the charge release of polymer materials.

Fig. R9 The charge release of a charged polymer and the experimental TSD current of a charged Teflon FEP. **a)** The charge release controlled by electron motions of a charged polymer. **b)** The charge release controlled by a synergistic effect of the molecular movements and the electron motions of a charged polymer. **c)** The experimental TSD current of a Teflon FEP (*J. Optoelectron. Adv. Mater.* 2006, 8, 962-966).

4. In the manuscript, the authors have stated that the surface area of the friction layers after treatment is increased. However, it cannot directly demonstrate that there is increased contact area since much surface area is not the real contact area.

Response: We thank the reviewer very much for the comments. The increased surface area of the friction layer is not strictly equal to the actual contact area, but it will indeed enlarge the real contact area and increase the triboelectric charge during the triboelectrification process. This has been verified by lots of previous studies (*Nano Lett.* 2012, 12, 3109-14; *J. Appl. Polym. Sci.* 2017, 135, 45674; *Energy Environ. Sci.* 2020, 13, 2178-2190). Maybe the description “The nanostructures fabricated on them can enhance the contact area (1.87 times and 1.96 times of the untreated Al foil and Kapton film respectively, Supplementary Fig. 3) during triboelectrification and improve TDNG’s electrical output performance” is not so rigorous. To be more rigorous, this sentence has been changed to “The nanostructures fabricated on them can enhance the effective surface area (1.87 times and 1.96 times of the untreated Al foil and Kapton film

respectively, Supplementary Fig. 3) to enhance the effective friction area and improve TDNG's electrical output performance”.

5. In the manuscript, the theoretical study in 2.1 has not been utilized to explain any experimental phenomena. Therefore, some discussions of figure 2 – 5 are suggested to be related to the theoretical study.

Response: We thank the reviewer very much for the comments. According to these important comments, we have carefully improved our work and revised the manuscript by combining theoretical study and experimental results. The detailed analyses and discussions are given below.

According to the theoretical study, assuming that the transferred charge equals to accumulated charge in the cooler friction layer, the relationship between transferred short-circuit charge quantities per cycle (CQC) of TDNG and temperature difference (ΔT) follows equation (36):

$$CQC = (-C_1 \frac{\Delta T + b}{1 - a} + C_2) e^{-SA t_0 A} \left[1 - \left(1 + \frac{d}{RA \epsilon_0 \epsilon_r} \int_0^t e^{\frac{1}{RA \epsilon_0 \epsilon_r} (\frac{d}{\epsilon_r} t + \int_0^t x(t) dt)} dt \right) e^{-\frac{1}{RA \epsilon_0 \epsilon_r} (\frac{d}{\epsilon_r} t + \int_0^t x(t) dt)} \right] \quad (36)$$

A nonlinear fit of CQC under different ΔT has been performed. As shown in **Fig. R10a**, the obtained material-related correction factor a equals to 0.1253, and the material-related correction factor b equals to 163.957. According to the theoretical study in Fig. 1d, when a equals to 0.1253, the corresponding thermal conductivity of the cooler friction layer can be deduced as $0.14 \text{ W} \cdot \text{m}^{-1} \text{K}^{-1}$, which is approximately the same as the thermal conductivity of Kapton ($\sim 0.12 \text{ W} \cdot \text{m}^{-1} \text{K}^{-1}$) that we used in experiments. Moreover, the obtained material-related correction factor b is also approximately in line with the result in the theoretical calculation (172.75). Furthermore, the obtained material-related correction factor C_1 equals to 0.0248, and C_2 equals to 3.1166. It conforms to the relationship between the transferred charge and T_h in the theoretical study, and it also conforms to the linear change in reference (*Adv. Mater.* 2019, 31, 1808197).

The above discussions are summarized into two sentences and added in the revised manuscript. In line 198, page 8, “Based on equation (3), a fitting analysis of CQC is

performed. As shown in Supplementary Fig. 10, the obtained material-related correction factor a equals to 0.1253, b equals to 163.957, C_1 equals to 0.0248, and C_2 equals to 3.1166. From these fitting parameters, the thermal conductivity of the cooler friction layer can be deduced as $0.14 \text{ W m}^{-1}\text{K}^{-1}$, which is approximately the same as the thermal conductivity of Kapton ($\sim 0.12 \text{ W m}^{-1}\text{K}^{-1}$) that we used in experiments.” is added. And in line 218, page 8, “With a fitting analysis, the obtained material-related correction factor a equals to 0.1253, b equals to 165.2682, C_1 equals to 0.0248, and C_2 equals to 4.256, which are basically the same as the values obtained from CQC (Supplementary Fig. 13).” is added.

Apart from the above numerical analyses, we also gave some qualitative analyses of some experimental data in the revised manuscript. For example, in line 186, page 7 of the revised manuscript, we gave the description “In accordance with the simulation, with the ΔT increasing from 0 K to 219 K, the output voltage and current of TDNG increase at first and then decrease. The largest output voltage and current can reach 858 V (measured by SR560 and a voltage divider, Supplementary Fig. 7) and $20 \mu\text{A}$ when ΔT equals to 145 K, as T_c keeps 299 K.”. In line 208, page 8 of the revised manuscript, we gave the description “As the schematic diagram of the TSD testing device (Supplementary Fig. 11) illustrates, when a charged Kapton film is heated by a heating system (the temperature rising curves of heating system in TSD testing is shown in Supplementary Fig. 12), the accumulated charges in Kapton will gradually escape from the potential well and a current peak is formed.”.

Fig. R10 The fitting analysis of the experimental results. The dots are measured values and the lines are fitting line that fitted by the theoretical model. **a)** The fitting analysis of the transferred short-circuit charge quantities per cycle (*CQC*). **b)** The fitting analysis of the surface potential.

REVIEWERS' COMMENTS

Reviewer #1 (Remarks to the Author):

Authors made a great effort to do some experiments raised by reviewers, which demonstrates the reliability of the results. I think it is a very interesting work and can be accepted for publication. One suggestion, the quality of the Figures should be improved, especially the captions in Figures are not large enough to be seen.

Reviewer #3 (Remarks to the Author):

Dear Authors,

The revised paper and responses have addressed our concerns. In particular, I appreciate the detailed responses and significant effort. I am happy to accept them.

Dear Dr. Ronghuan Zhang and reviewers,

We are very grateful for your recognition and consideration of publishing our manuscript. According to the suggestion from reviewer, we have improved the quality of the figures and enlarged the captions in figures for better readability. The revised manuscript and supplementary information are uploaded online. Thanks.

With best regards,

Sincerely yours,

Yong Qin

Professor, Lanzhou University

Response to the reviewers' comments

Response to reviewer 1

Comments:

Authors made a great effort to do some experiments raised by reviewers, which demonstrates the reliability of the results. I think it is a very interesting work and can be accepted for publication. One suggestion, the quality of the Figures should be improved, especially the captions in Figures are not large enough to be seen.

Response: We appreciate the reviewer's very positive and valuable comments during the review process. The reviewer's comments and suggestions help us to improve the quality of our work. Besides, according to your suggestion, the quality of the figures has been improved and the captions in figures have been enlarged for better readability. Thanks.

Response to reviewer 3

Comments:

Dear Authors, the revised paper and responses have addressed our concerns. In particular, I appreciate the detailed responses and significant effort. I am happy to accept them.

Response: We appreciate the reviewer very much for the positive evaluation and the valuable comments in the review processes. According to these important comments, the quality of our work is improved. Thanks.